# Cognitive reserve involves decision making and is associated with left parietal and hippocampal hypertrophy in neurodegeneration

Lorna Le Stanc [1,2,3,4], Marine Lunven[1,2,3], Maria Giavazzi[1,2,3], Agnès Sliwinski[1,2,3,5], Katia Youssov[1,2,3,5], Anne-Catherine Bachoud-Lévi [1,2,3,5] & Charlotte Jacquemot [1,2,3] ✉

Cognitive reserve is the ability to actively cope with brain deterioration and delay cognitive decline in neurodegenerative diseases. It operates by optimizing performance through differential recruitment of brain networks or alternative cognitive strategies. We investigated cognitive reserve using Huntington's disease (HD) as a genetic model of neurodegeneration to compare premanifest HD, manifest HD, and controls. Contrary to manifest HD, premanifest HD behave as controls despite neurodegeneration. By decomposing the cognitive processes underlying decision making, drift diffusion models revealed a response profile that differs progressively from controls to premanifest and manifest HD. Here, we show that cognitive reserve in premanifest HD is supported by an increased rate of evidence accumulation compensating for the abnormal increase in the amount of evidence needed to make a decision. This higher rate is associated with left superior parietal and hippocampal hypertrophy, and exhibits a bell shape over the course of disease progression, characteristic of compensation.

Neurodegenerative diseases affect brain parts and functions at variable degrees and at different stages over the course of the disease, and eventually precipitate brain atrophy that precedes intellectual deterioration[1]. In general, patient's normal behaviour is maintained until the neuropathological damage surpasses the adaptive capabilities of the brain leading to the appearance of clinical symptoms[2–4]. The concept of "reserve" refers to this capacity of the brain to resist neuropathological changes and preserve cognitive functioning[5]. Reserve is thought to rely on brain reserve and cognitive reserve. While brain reserve relies purely on quantitative aspects of the brain such as brain size for example, cognitive reserve reflects the brain's capabilities to optimize and develop alternative cognitive strategies to actively preserve cognitive functions. Cognitive reserve depends on patient's lifetime intellectual activities and environmental factors. It relies on two concepts: neural reserve and neural compensation. The brain can either increase the efficiency of an existing yet deteriorating network (neural reserve) and/or recruit other regions upon performing a task (neural compensation)[3,6]; i.e., some cognitive functions may compensate for others that were impacted at earlier stages.

Assessing cognitive dysfunction concealed by cognitive reserve requires disentangling them from each other to identify each separately. The difficulty lies on the fact that most neurodegenerative diseases remain undiagnosed until clinical manifestation, a point at which it is too late to study cognitive reserve since its mechanisms are no longer effective. Inherited neurodegenerative diseases provide a promising way of overcoming this limitation. Huntington's disease is an inherited, monogenetic (expanded CAG repeat in the Huntingtin gene), dominant, and fully penetrant neurodegenerative disease[7]. Individuals who carry >40 CAG repeats will develop the disease[8]. This specificity allows to identify premanifest Huntington's disease gene carriers (preHDs) in whom striatal atrophy has already settled, while the disease clinical onset and its related motor, cognitive, and psychiatric deterioration have not yet appeared[9]. Therefore, Huntington's disease is a particularly well-suited

[1]Département d'Études Cognitives, École Normale Supérieure-PSL, Paris, France. [2]Institut Mondor de Recherche Biomédicale, Inserm U955, Equipe E01 Neuropsychologie Interventionnelle, Créteil, France. [3]Université Paris-Est Créteil, Faculté de Santé, Créteil, France. [4]Université Paris Cité, LaPsyDÉ, CNRS, F-75005 Paris, France. [5]AP-HP, Centre de Référence Maladie de Huntington, Service de Neurologie, Hôpital Henri Mondor-Albert Chenevier, Créteil, France. ✉e-mail: charlotte.jacquemot@ens.psl.eu

neurodegenerative model for studying compensation mechanisms from genetic diagnosis to onset of overt clinical manifestations[4,7,10].

To study the mechanisms of cognitive reserve separately from cognitive dysfunction[11–14], we selected a cognitive task that is impaired in the early stages of the disease (no longer effective cognitive reserve) but still normal in presymptomatic subjects (effective cognitive reserve). We used a language discrimination task as language is one of the first cognitive function to decline in Huntington Disease[14–19], with normal or near-normal performance in preHDs and abnormal performance in earlyHDs, suggesting that language performance would be a reliable measure to study cognitive reserve. We analyzed the results of the language task—deciding whether two items are similar or not—through Drift diffusion models (DDMs)[20,21]. These models enable us to evaluate separately the cognitive parameters involved in the discrimination task: accumulation over time of sensory evidence at a certain rate up to a response threshold that triggers the motor response to indicate which of the two alternatives to pick (similar or not). The hypothesis is that in preHD, one of the impaired cognitive parameters will be compensated for by another cognitive parameter, resulting in normal behavioural performance, whereas in earlyHD, the compensatory cognitive parameter will no longer be effective, resulting in a behavioural deficit.

We thus analyzed behavioural data of the language discrimination task and used DDMs to separate subclinical deficits from cognitive reserve mechanisms in preHDs. In order to explore in depth the progression of the cognitive reserve mechanisms over the course of the disease, we divided preHDs into groups according to the time remaining until the predicted age-at-onset. Additionally, we used neuroanatomical data to explore the correlation between cognitive reserve mechanisms and brain structural adaptations to such mechanisms. Behavioural and neuroanatomical analyses first confirmed that preHDs behave similarly to controls, despite their incipient striatal atrophy, confirming that cognitive reserve is effective at the premanifest stage of the disease. DDMs revealed that cognitive reserve in preHDs relies on decision making, with an increased drift rate which compensate for a higher response threshold both observed in earlyHDs and preHDs. The cognitive underpinnings of cognitive reserve correlate with left superior parietal hypertrophy and hippocampal hypertrophy, revealing a neural network that compensates for brain degeneration.

## Results

We applied DDMs to 93 participants of whom 20 where premanifest Huntington's disease mutation carriers (preHDs) without overt cognitive symptoms, 28 were early-stage Huntington's disease patients (earlyHDs), and 45 were controls. The two groups of mutation carriers (earlyHDs and preHDs) were distinguished from each other following the clinical evaluation based on the Unified Huntington's Disease Rating Scale[22]. The demographic and clinical characteristics of participants are summarized in Supplementary Table 1.

We evaluated participants' cognitive functions using three tests of the Unified Huntington's Disease Rating Scale: verbal fluency test, Stroop test, and Symbol digit modalities test[22]. Participants were further evaluated using forward digit span, category fluency, and trail making test (TMT)[23]. In addition, participants performed a simple AX auditory language discrimination task in which they were asked whether two pseudowords were identical or different (Fig. 1a). We calculated the mean accuracy and the mean response time for each participant. We hypothesized that preHDs, relying on their cognitive reserve mechanisms, should show little deficit, if any, whereas earlyHDs should show an overt deficit, assuming that their compensation mechanisms would no longer be effective.

For each of the cognitive and behavioural tests, we performed one-way ANOVA, with age as a covariate, to study the difference between groups (controls, preHDs, earlyHDs).

### PreHDs presented a normal clinical and behavioural profile

Analyses of cognitive tests showed a significant main effect of group in each test. Tukey's post-hoc analyses revealed that the earlyHDs performed worse in all tests, compared with the preHDs and controls (all $p < 0.05$). On the other hand, the preHDs' performances were statistically similar to those of the controls (all $p > 0.05$) (Supplementary Table 2).

As for the mean accuracy and mean response time of the language task, analyses showed a significant main effect of group (accuracy: $F(2,90) = 20.93$, $p < 0.001$, $\eta^2 = 0.32$; response time: $F(2,90) = 41.06$, $p < 0.001$, $\eta^2 = 0.48$). Tukey's post-hoc analyses revealed that the earlyHDs were less accurate and slower than controls and preHDs. In contrast, the preHDs were not statistically different from the controls (Supplementary Table 3).

### PreHDs presented neural atrophy

Forty-six three-dimensional, T1-weighted, structural brain MRI scans were obtained within about 3 months from behavioural data acquisition in mutation carriers (20 preHDs and 26 earlyHDs). They were compared with 30 scans from a cohort of external healthy participants who did not perform the language task and the cognitive battery (imaging controls). The imaging controls had no previous or current neurological or psychiatric history and their anatomical MRI was checked by a neuroradiologist for any abnormalities. They were matched with the mutation carriers for age and sex (46.1 ± 13.9 years old, 15 females).

We first compared the volumes of subcortical structures (striatum, pallidum, thalamus, hippocampus, and amygdala) between the mutation carriers and imaging controls. We ran mixed ANOVA with subcortical volumes normalized to the total intracranial volume as the dependent variable, group as a between factor, subcortical structure as a within factor, and age as a covariate. There was an interaction between subcortical structure and group ($F(8,372) = 34.32$, $p < 0.001$, $\eta_p^2 = 0.03$). Tukey's post-hoc showed that earlyHDs had lower grey matter volumes than both imaging controls and preHDs, in the striatum (all $p < 0.001$), pallidum ($p = 0.0510$, least), and thalamus (all $p < 0.01$) (Fig. 2, Supplementary Table 4). In addition, preHDs displayed striatal atrophy upon comparing them with the imaging controls ($p < 0.001$). Overall, there was no statistical evidence for a difference between groups concerning the volumes of the hippocampus and amygdala (all $p > 0.05$).

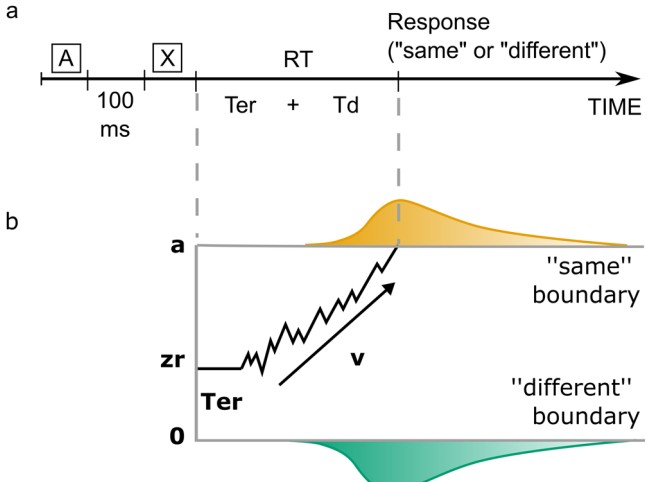

**Fig. 1 | Discrimination task and hierarchical drift diffusion model. a** Participants heard two pseudowords (A and X) separated by a 100 ms time interval, and had to decide whether they were identical or not. The response time (RT) is the sum of the non-decision time (Ter) and the decision processes (Td): RT = Ter + Td. **b** Example of the trajectory of the drift diffusion model for a "same" trial in which the correct response was delivered. Two decision boundaries (**a** and 0) represent the "same" and "different" decisions, respectively. The drift rate, v, represents the rate of evidence accumulation. The diffusion process starts between the two boundaries at zr (0.5a, if not biased toward one of the alternatives) and continues until it reaches one of the two boundaries. The predicted response time is the sum of the durations of the diffusion process called decision time and the one called non-decision time, which encompasses stimulus pre-processing and motor planning and execution.

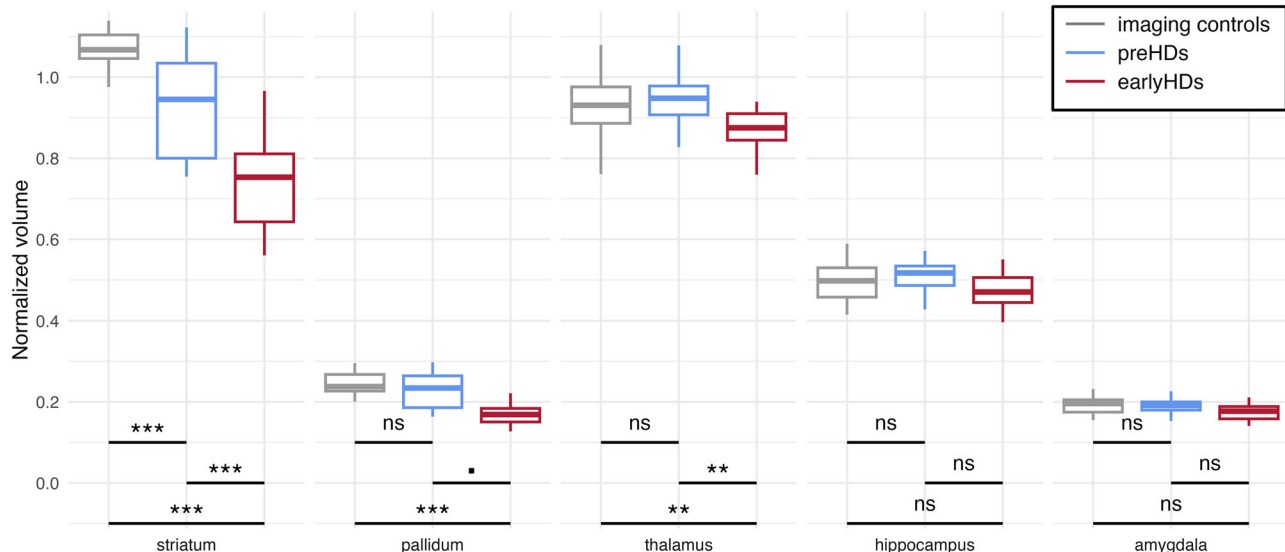

**Fig. 2 | Neuroanatomical differences between groups.** Boxplots of the subcortical volumes differences between groups. For representational purposes, volumes normalized to the total intracranial volume (tiv) were multiplied by 100 for all structures. In boxplots, the middle hinge corresponds to the median, the lower and upper hinges correspond respectively to the first and third quartiles. $***p < 0.001$, $**p < 0.01$, $\blacksquare p = 0.0510$, ns: non-significant. Imaging controls are represented in grey ($n = 30$), premanifest participants (preHDs) in blue ($n = 20$), and early-stage Huntington's disease patients (earlyHDs) in red ($n = 26$).

The following step was to study the differences in cortical thickness. EarlyHDs showed cortical thinning in the left angular gyrus, the left occipital superior cortex, the right caudal part of the middle frontal cortex, and the right lateral occipital lobe upon comparing them with imaging controls (Supplementary Fig. 1a and Supplementary Table 5). EarlyHDs also had a thinner right lateral occipital cortex than that of preHDs (Supplementary Fig. 1b and Supplementary Table 5). Cortical thickness was similar in preHDs and imaging controls.

## DDM revealed cognitive reserve

DDMs assume that, in order to make a decision between two alternatives, one needs to extract sensory information from the stimulus. This information is accumulated, over the decision-making process, at a certain speed, called the drift rate (v), up to a response threshold (a) that triggers the motor response. Evidence accumulation is therefore a time-consuming and noisy process requiring multiple samples to extract information from the stimulus before enough evidence is collected to make a decision. The time required for non-decision processes, such as stimulus processing, motor preparation, and execution, is captured in the non-decision time (Ter). The a priori bias towards one of the alternatives is called the relative bias (zr)[24](Fig. 1b). For a given participant, the four parameters (v, a, Ter, and zr) were obtained by fitting the distribution of responses ("same" or "different") and their corresponding response time in each trial. In our task, the participant would hear two pseudowords at each trial. First, the acoustic information is encoded (stimulus processing). Then the two pseudo-words are compared at a certain speed (drift rate of evidence accumulation). According to the participant's conservatism, the amount of evidence required to choose "same" or "different" is more or less elevated (response threshold). Once the decision is taken, the participant presses the corresponding key (motor preparation and execution). For each DDM parameter, we ran ANOVA to study the difference between groups after introducing age as a covariate.

Analyses showed a significant difference between groups in terms of the response threshold ($F(2,90) = 4.79$, $p < 0.05$, $\eta^2 = 0.10$) and the drift rate ($F(2,90) = 5.60$, $p < 0.01$, $\eta^2 = 0.34$), unlike non-decision time ($F(2,90) = 0.13$, $p = 0.88$, $\eta^2 = 0.003$) and relative bias ($F(2,90) = 0.24$, $p = 0.79$, $\eta^2 = 0.005$). Tukey's post-hoc showed progression of the response threshold from the lowest in controls to the highest in earlyHDs, with the values of the preHDs in between. This suggests an increase of the response threshold over the course of the disease (Supplementary Table 3, Fig. 3a). On the contrary, the drift rate showed a bell-shaped pattern, a recognizable signature of compensation (Supplementary Table 3, Fig. 3b). PreHDs and controls presented a higher drift rate than earlyHDs. Although not significant, preHDs tended to have a higher drift rate compared to controls.

To further investigate this pattern, we split the preHDs into three groups according to their time to predicted age-at-onset[8]. Determination of disease onset in Huntington's disease is made by clinical experience, but the conversion is a progressive process which makes it difficult to determine the exact moment of motor onset of HD[25]. PreHDs were thus stratified into three groups, far, middle and close to the onset of the disease, according to the time remaining until the predicted onset of the disease: close to onset with predicted onset within a year ($N = 4$), far to disease onset with a predicted onset $\geq 10$ years ($N = 7$) and middle onset with a predicted onset between 1 and 10 years ($N = 9$)[26,27]. Conducting ANOVA analyses with these new groups and age as a covariate confirmed the bell-shaped pattern. Analyses showed a main effect of group ($F(4,88) = 15.29$, $p < 0.001$, $\eta^2 = 0.41$). Tukey's post-hoc showed that preHDs far from onset had a statistically similar drift to that of controls, whereas the preHDs of the middle group had a drift rate superior to those of controls, preHDs far from onset, and earlyHDs. Finally, preHDs close to onset had a drift rate statistically similar to that of controls, with a value between that of middle preHDs and that of earlyHDs (Fig. 3d, Supplementary Table 6).

The participant's behavioural performance is the result of the drift rate and response threshold parameters. Drift rate appears to show a compensatory pattern (faster rate of evidence accumulation) while response threshold shows a deteriorative pattern (higher response threshold) throughout all stages of the disease. We went further and investigated the relationship between drift rate and response threshold. We fitted a linear model with the drift rate as a dependent variable, response threshold and group as predictors, and age as a covariate. This showed that an increase in response threshold was associated with an increase in drift rate. The increase was similar in controls and earlyHDs, and sharper in preHDs (Supplementary Table 7, Fig. 3e), meaning that for preHDs the drift rate might compensate for the increasing threshold allowing to preserve behavioural performances. This relationship between parameters was not induced by contamination between them as they were all not correlated (all $p > 0.05$)[28].

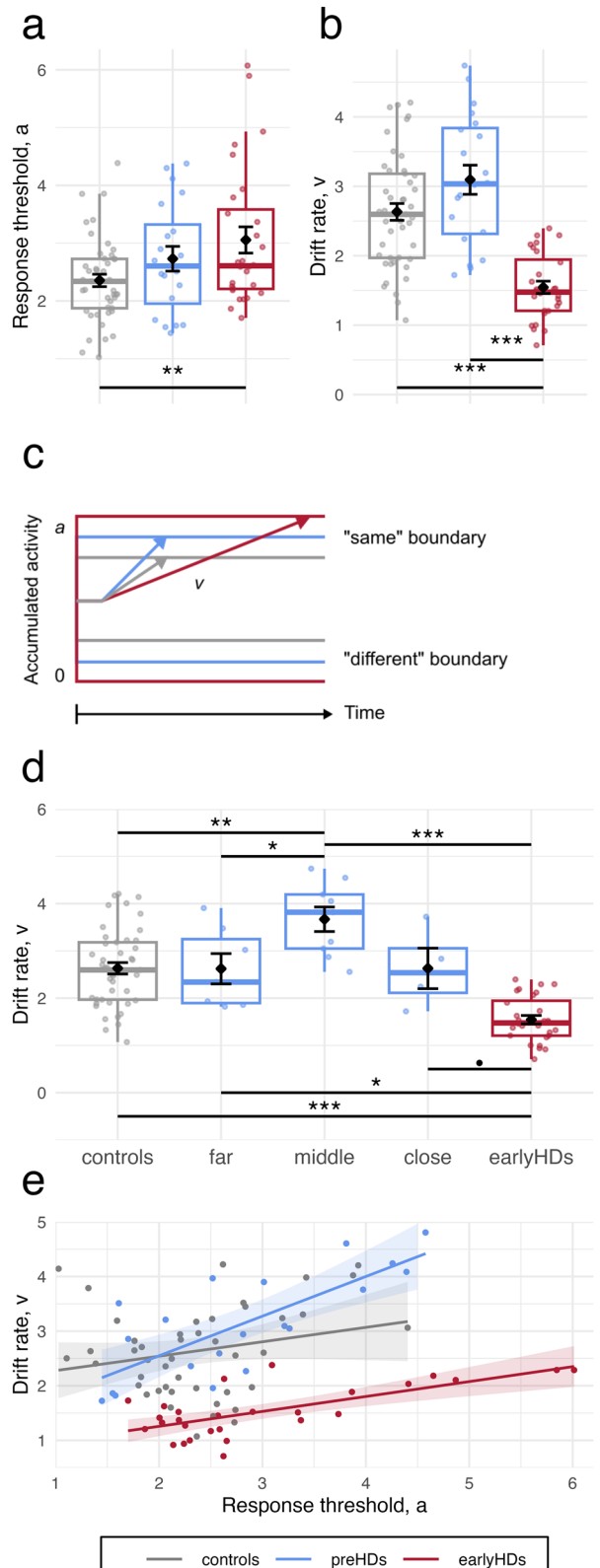

**Fig. 3 | Results of DDM analyses.** Boxplots of (**a**) Response threshold a and (**b**) drift rate v of each group. **c** Schematic representation of model parameters of each group. **d** Relationship between the drift rate (y-axis) and time to predicted age-at-onset (x-axis). **e** Relationship between the drift rate (y-axis) and the response threshold (x-axis). Points represent individual values, lines and shades around them represent the linear fit and the confidence interval, respectively. Controls are represented in grey (n = 45), premanifest participants (preHDs) in blue (n = 20, n = 7 far, n = 9 middle, n = 4 close to onset), and early-stage Huntington's disease patients (earlyHDs) in red (n = 28). In boxplots in (**a**), (**b**), and (**d**), the middle hinge corresponds to the median, the lower and upper hinges correspond respectively to the first and third quartiles.

controlled for the effect of the latter using the total intracranial volume as a proxy in our analyses[6].

**Drift rate is associated with hippocampus volume in preHDs.** First, we explored the relationship between the drift rate and subcortical structures volumes normalized to the total intracranial volume (striatum, pallidum, thalamus, amygdala, and hippocampus) using a linear model with the drift rate as the dependent variable, subcortical structures and disease stage as the predictive variable, and age as a covariate. There was no significant effect of any of the structures (all $p > 0.05$), however, there was an interaction between the disease stage and the hippocampus volume ($F(1,33) = 13.35$, $p < 0.001$, $\eta^2_p = 0.29$). A higher volume of the hippocampus was associated with a higher drift rate only in preHDs ($\beta = 2246 \pm 555$, 95%CI [1118, 3374], $t(33) = 4.05$, $p < 0.001$) and not in earlyHDs ($\beta = -160.3 \pm 351$, 95%CI [-874, 553], $t(33) = -0.46$, $p = 0.88$) (Fig. 4a, c). When we looked at that closely with follow up analyses, preHDs middle group had a bigger hippocampus volume than that of earlyHDs ($\beta = 0.0005$, 95%CI [0.00007, 0.0010], $p < 0.05$) (Fig. 4f).

**Drift rate is associated with left superior parietal thickness in mutation carriers.** Second, we explored the relationship of the drift rate with the cortical thickness while controlling for the total intracranial volume. Lower drift rates were associated with a thinner cortex in clusters located in the left superior parietal cortex and the right superior temporal gyrus (all $p < 0.05$), independently of the disease stage (no significant clusters with interaction, all $p > 0.05$) (Fig. 4b, d, e). In these two significant cortical clusters, earlyHDs had a thinner cortex compared with imaging controls ($p < 0.001$). There was no significant difference between preHDs and imaging controls (all $p > 0.05$), though preHDs tended to have a thicker cortex especially in the left superior parietal cortex (one sided $t$-test $t(41) = 1.71$, $p = 0.047$, $d = 0.49$).

**Left superior parietal thickness presents a bell shape pattern related to age-at-onset.** Post-hoc analyses including the sub-groups of preHDs revealed that earlyHDs had a thinner right superior temporal cortex compared with all sub-groups of preHDs ($p < 0.01$, least). There were no other significant differences between any of the sub-groups of preHDs and imaging controls (Fig. 4h). In the superior parietal cortex, there was no significant difference between preHDs and imaging controls ($p > 0.05$). Notwithstanding, preHDs middle group had a significantly thicker cortex compared with imaging controls, earlyHDs, and preHDs close to onset ($p < 0.05$, <0.001, and <0.05, respectively) in the left superior parietal cortex (Fig. 4g). In the latter cluster, preHDs close to onset had similar cortical thinness to that of earlyHDs ($p > 0.05$).

One should highlight here that the hippocampus volume and the cortical thickness of the two clusters were not correlated with any of the clinical and cognitive tests. In another word, there is no statistical evidence that the observed changes in brain anatomy were not specifically related to the increase in drift rate.

**Response threshold is correlated with hippocampus volume in preHDs.** Regarding the response threshold, there was an interaction between the disease stage and the hippocampus volume ($F(1,33) = 4.21$,

**Cognitive reserve is correlated with brain structures in mutation carriers**

To explore the neuro-correlates of cognitive reserve, we investigated the relationship between the drift rate and neuroanatomical data among mutation carriers (preHDs and earlyHDs). To make sure that we were looking at the effect of cognitive reserve rather than the brain reserve, we

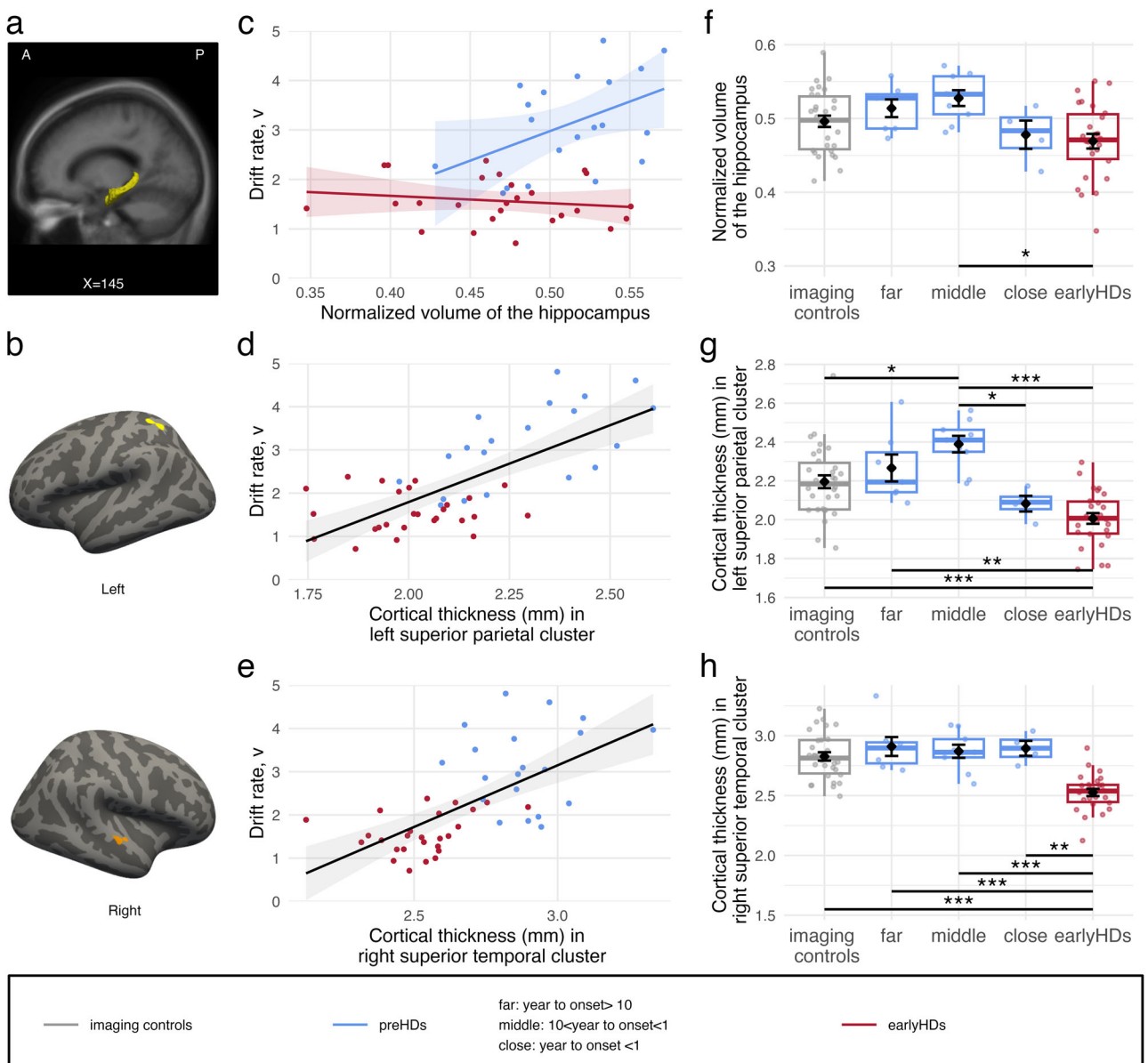

**Fig. 4 | Relationship between brain structure and drift rate. a** Hippocampus (yellow). X is MNI coordinate. A anterior, P posterior. **b** Cortical maps of significant clusters with significant relationship between the cortical thickness and the drift rate. Light grey represents gyrus and dark grey represents sulcus. Yellow: left superior parietal cluster (152.71 mm2, MNI coordinates: [−34, −55, 59]); Orange: right superior temporal cluster (112.72 mm2, MNI coordinates: [63, −16, −1]). **c** Relationship between the hippocampus volume normalized to the total intracranial volume (tiv) (x-axis) and the drift rate (y-axis). For representational purposes, the x-axis is multiplied by 100. **d** Relationship between the mean cortical thickness in the yellow cluster (panel **b**) (x-axis) and the drift rate (y-axis). **e** Relationship between the mean cortical thickness in the orange cluster (**b**) (x-axis) and the drift rate (y-axis). **f** Boxplots of the normalized hippocampus volume. **g** Boxplots of the mean cortical thickness in the yellow cluster. **h** Boxplots of the mean cortical thickness in the orange cluster. *$p < 0.05$, **$p < 0.01$, ***$p < 0.001$. Imaging controls are represented in grey ($n = 30$), premanifest participants (preHDs) in blue ($n = 20$, $n = 7$ far, $n = 9$ middle, $n = 4$ close to onset), and early-stage Huntington's disease patients (earlyHDs) in red ($n = 28$). In **c**, **d**, and **e**, points represent individual values, whereas lines and shades around them represent the linear fit and the confidence interval, respectively.

$p < 0.05$, $\eta^2_P = 0.01$): a higher volume of the hippocampus predicted a higher response threshold in preHDs ($\beta = 2163.5 \pm 892$, 95%CI [349, 3978], $t(33) = 2.43$, $p < 0.05$), but not in earlyHDs ($\beta = −10.9 \pm 564$, 95% CI [−1158, 1137], $t(33) = −0.02$, $p = 0.99$). There were no significant relationships between the cortical neuroanatomical structure and the response threshold nor significant interaction with the disease stage (all $p > 0.05$).

## Discussion
In this study, our aim was to identify the mechanism underlying cognitive reserve in neurodegenerative diseases using Huntington's disease as a model. We used DDMs to gain insight into the decision-making processes involved in an language discrimination task we designed (Fig. 1). As previously reported, analyses of clinical cognitive assessment and behavioural performances showed that preHDs performed as well as healthy participants (Fig. 3a, b) despite displaying incipient atrophy of the striatum (Fig. 2)[1,29,30] and cerebral functional changes[31–35]. However, analyses of DDMs parameters revealed a different profile in preHDs with a substantial increase in response threshold predicting a faster drift rate of evidence accumulation (Fig. 3c–e). We hypothesized that the higher drift rate in preHDs is a compensatory mechanism that preserves normal accuracy and response times (Fig. 3e). The association between the increase in response

threshold and the increase in drift rate of accumulation is consistent with compensation between these two processes. This compensation was further evidenced by the bell-shaped pattern of drift rate seen upon dividing the preHDs into three groups according to their time to predicted age-at-onset, where the middle group scored the highest (Fig. 3d). In contrast, earlyHDs displayed impairment in clinical cognitive assessment, behavioural performances, and DDMs analyses, suggesting that compensatory mechanisms were absent or insufficient to counterbalance the decline as the disease progresses and the pathological load increases, leading to observable cognitive impairments[2,4] in these individuals. In preHDs, the spared thalamus (Fig. 2), the relationships between the drift rate and both the increased volume of the hippocampus (Fig. 4c) and the hypertrophy of the superior parietal cortex (Fig. 4d) suggest that the compensatory mechanism might be related to the attentional network[36–38].

In our task, the response threshold increased across disease stages suggesting a gradient of impairment proportional to disease progression (Fig. 3c). A higher response threshold leads to a slower decision-making process. In healthy participants, Forstmann et al. (2010)[39] showed that the flexible variation of the DDMs response threshold was dependent on the strength of the connections between the cortex and striatum that inhibit the subthalamic nucleus. This indirect pathway from the striatum to the thalamus through the external globus pallidus and subthalamic nucleus was much more affected in earlyHDs than in preHDs[40]. Disruption of that indirect pathway, and a decrease in the number of white matter fibres extending between the striatum and the cortex[41,42] in Huntington's disease should decrease the inhibition of the subthalamic nucleus and lead to less impulsive choices and an increase in response threshold[43]. This could explain why we could not find neuroanatomical correlates of the response threshold since we explored only brain volume and cortical thickness. In line with recent research on the role of white matter in adaptation to neurodegeneration[44], examining the white matter integrity and network connectivity would be warranted to complement this work.

Provided that earlyHDs' other cognitive disabilities are not too severe, longer decision times should improve their accuracy (speed-accuracy tradeoff)[45]. Yet, this is not the profile of responses we observed as earlyHDs are less accurate and slower than controls, showing a performance impairment compared to controls. Their slower rates of evidence accumulation, brought by poor quality of evidence extracted from short-term storage (lower drift rate), increased the number of errors they made and reduced their performance in perceptual decision-making. Based on such findings, we built our hypothesis that preHDs presented a compensatory rise in drift rate that helped them maintain their normal behaviour despite the parallel rise in response threshold (Fig. 3e). The stronger relationship between drift rate and response threshold in preHDs compared with their counterparts, together with the bell-shaped pattern it gave with preHDs subgroups (Fig. 3d), and the inability of earlyHDs to maintain normal behavioural performances are consistent with models assuming that compensation mechanisms become less effective over the disease progression[2,4]. The drift rate may constitute a measurable cognitive marker of compensation.

Drift rate is linked to attention in both healthy subjects[21] and patients with hyperactivity disorder[46]. Individuals with higher attentional capacities accumulate evidence faster[47]. We used MRI imaging to reveal drift rate-related compensatory mechanisms and saw that the drift rate correlated positively with cortical thickness in the left superior parietal and the right superior temporal cortices, which are associated with a better ability to sustain attention (Fig. 4d, e)[36–38]. Within this attentional network, the hippocampus plays a role in maintaining high-resolution representations in working memory when a complex and precise representation is required[48], especially in online perception[49]. The content of working memory automatically modulates attention by gating the information matching its content into awareness[50]. In preHDs, a larger hippocampal volume predicted a higher drift rate (Fig. 4c). This raises the possibility that the hippocampus contributes to tune their attention to relevant stimulus features (fine-grained presentation of pseudowords in our task), which consequently

increased information extraction. Obviously, the structural modification (hypertrophy) (Fig. 4a, d), observed in preHDs, has developed over time with daily use of this mechanism, beyond our simple task. In contrast, hippocampal volume was not related to drift rate in earlyHDs suggesting that attentional tuning was no longer working at this stage. The inability of earlyHDs to recruit sufficient additional attentional resources is consistent with their brain atrophy and the pattern of attentional impairment observed in this disease. In our cohort, two key components of the attentional network[36,51], i.e. the right caudal part of the middle frontal cortex and the thalamus, were atrophied in earlyHDs and spared in preHDs (Fig. 2, Supplementary Fig.1). In line with previous literature, preHDs were minimally affected in this domain, whereas earlyHDs presented a wide range of attentional deficits (e.g. regarding sustained attention[52]). Moreover, the atrophy of the left angular gyrus, a key structure involved in phonological discrimination[53], seen in earlyHDs was previously reported[54]. This might also have prevented them from biasing their attention to fine-grained phonological features of our task.

Cognitive reserve that operates through active compensation mechanisms may depend on either an increase in the activity of a deficient network (neural reserve) or the recruitment of alternative networks with available resources (neural compensation)[6]. Both have been observed in preHDs where functional imaging studies showed changes in BOLD responses in task-dependent regions despite similar behavioural performances to that of controls[4,32,33]. Changes in connectivity, structure, or activation can provide information about the link between the disease and neural reorganization. However, this link might be pathological rather than compensatory if it is not correlated with improved performances[2,3]. Here, we observed left superior parietal cortex hypertrophy in preHDs (Fig. 4d). Such an increase in cortical thickness, supposedly caused by hyperactivation, was associated with better performances (shorter response times, better accuracy, and higher drift rates) (Supplementary Fig. 2, Supplementary Table 8, Fig. 4d, e), supporting the hypothesis of a successful compensation, as previously reported in motor learning[31]. The post-hoc analysis showed that only preHDs middle group had this cortical hypertrophy, not preHDs far from and close to predicted age-at-onset; a bell-shaped pattern in favour of a compensatory mechanism emerging at a certain point and failing as the pathological load increases. Although these findings would need to be replicated in a larger cohort, they are consistent with the results of previous studies on Alzheimer's disease[55] and Huntington's disease[56] which reported a preclinical stage of hypertrophy and increased functional connectivity between the left caudate nucleus and parietal lobe preceding atrophy in symptomatic patients[34]. This may reflect an experience-dependent increase in neural volume[57,58] as an attempt to compensate for the dysregulation of the striatal network.

Overall, whether the compensatory mechanism is induced by attention modulation which refers to neural reserve, or attention recruitment which refers to neural compensation, is not resolved here. The fact that preHDs did not show cortical atrophy of the attentional system (see Supplementary Fig. 1, Supplementary Table 5) indicates that this network was not a priori affected. Future research replicating the results using functional MRI data and tasks aimed at testing attentional capacities in HD mutation carriers would allow to confirm the overactivation or additional recruitment of the attentional network and to capture the dynamic nature of cognitive reserve. It is worth mentioning here that our imaging results corresponded to cognitive reserve rather than to brain reserve since we took total intracranial volume into account in cortical thickness analyses and normalized volumes in subcortical analyses.

Despite the study small cohort size, DDMs detected differences between preHDs and controls and identified cognitive processes that might underlie compensatory mechanisms in a discrimination task. We have shown that DDMs, combined with a language task, had an added value over classical neuropsychological tests, which rarely detect differences in preHDs, unless hundreds of participants are recruited[1,59]. Choosing a language task was motivated by studies showing the sensitivity of language in detecting subtle disorders in small cohorts of preHDs[14–18]. The duration of

the task and its simplicity (pseudoword discrimination) made it easily adaptable and transferable to other languages and other brain pathologies. Showing the generality of the drift rate as a marker of compensation in preHDs, i.e., of the increase in attention allocation as a compensatory mechanism, would require assessing other cognitive domains. Structural imaging provides a hint into the attentional network. Yet, it does not fully capture and take into account the dynamic nature of cognitive reserve as a process and does not allow discriminating between neural reserve or neural compensation for example. Studying the functional correlates of the drift rate in Huntington's disease would provide a greater understanding of online allocation of attentional resources as a compensatory mechanism[4,60]. These finding suggest that focusing on the role of the superior parietal cortex and hippocampus is promising in Huntington's disease, albeit needs to be replicated in other neurodegenerative diseases. Strengthening the cognitive mechanisms that compensate for the degeneration of the brain in neurodegenerative diseases might allow to delay the onset and progression of these diseases and offer a promising line of treatment for neurodegenerative diseases.

## Methods

### Participants

We recruited native French-speaking adults who were Huntington's disease mutation carriers in addition to a group of healthy participants as controls. Mutation carriers were either at an early-stage of Huntington's disease (earlyHDs; classified stages I and II based on total functional capacity score of the Unified Huntington's Disease Rating Scale), or were at the pre-manifest stage (preHDs; total functional capacity score of 13 and total motor score of <5[1] with no overt cognitive deficits[9]). Clinical data were collected using paper-and-pencil tests. The recruited controls were matched with the earlyHDs and preHDs groups for demographic variables, such as sex, handedness, years of education, and age (all $p > 0.05$). The two mutation carrier groups were further matched for demographic variables (all $p > 0.05$) except for age ($p < 0.05$). The participants had no neurological or psychiatric disorders other than Huntington's disease in the mutation carriers.

This study was conducted in accordance with the Declaration of Helsinki (2008). All ethical regulations relevant to human research participants were followed. Participants were recruited between December 2013 and July 2017 from a clinical biomarker study (NCT01412125) in outpatients approved by the ethics committee of *Henri Mondor Hospital* (Créteil, France). The sample size was estimated on the basis of previous work[61]. Participants inclusion ended when 45 valid brain MRI scans had been obtained from mutation carriers. All participants gave written informed consent.

### Experimental design

We designed a simple AX auditory language discrimination task that relied on an automatic linguistic process rather than explicit learning[62,63]. We focused on a low-level language component, phonological processing, in a task optimized to induce performance differences between controls and HD mutation carriers. In addition to being automatic[64,65] phonological processing has a distinct brain signature from non-phonological processing[53].

The discrimination task consisted in distinguishing a pair of pseudowords read 100 ms apart (Fig. 1a). The two pseudowords were identical in a half and differed by a single consonant (e.g. /tiplysk/ and /tipʁysk/) in the other half. The location of the consonant that differed varied between the trials ($N = 216$ trials) to impede expectation. A female native French speaker (the last author) pronounced the pseudowords for the recording, with each pseudoword lasting $1030 \pm 165$ ms. Once the participant gave a response, another trial would start 1000 ms after. The task lasted <10 min in total. Except for the training session, trials were randomized into two blocks separated by a break.

The participants were asked to sit in front of an Apple MacBook Pro, in a quiet room, wearing headphones tuned to ensure hearing comfort and had to press P for "*pareil*" (the French word for "same") or D for "different", on an AZERTY keyboard. Participants were informed that their accuracy and

response time would be recorded, and were advised to answer as accurately and quickly as possible. The experiment was run under PsyScope X[66].

### Analysis of clinical assessment, response time, and accuracy

We analyzed the effect of group on the results of cognitive tests (forward digit span, category fluency, trail making test (TMT), Unified Huntington's Disease Rating Scale cognitive scores) and language discrimination task (mean accuracy and mean response time) using one-way ANOVA with group as a between-participants factor. Age was added as a covariate to take into account the normal difference between preHDs and earlyHDs brought along by the natural progression of the disease. Whenever ANOVA analyses showed a significant effect of group, pairwise comparison using Tukey's post-hoc analyses were undertaken (three pairs) to find out which group means were different.

For accuracy and response time, the training trials were not included in the analyses. However, trials in which participants withdrew temporarily from the experiment and/or answered before the end of the trial presentation were removed, which resulted in losing 0.06% of data. Accuracy analyses were run on the remaining trials. Response time analyses were run on correct-answer trials (3.9% loss) lasting >150 ms (8.1% loss).

### Model fit and selection

Bayesian hierarchical DDM[67] is currently the most efficient method for dealing with a small number of observations[68], hence its use in our work. It assumes that individual parameter estimates are random samples of group-level distributions. Data were cleaned as in behavioural analyses, albeit Bayesian hierarchical DDMs used both correct and incorrect responses and response time. We assumed the same absolute drift rate value for both answers ("same" and "different"), a necessary hypothesis to estimate a possible relative bias toward one of the answers.

We tested two variants of the Bayesian hierarchical DDMs, full versus parsimonious. In the full model, each parameter had three group-level distributions, corresponding to our three groups (controls, preHDs, earlyHDs). The parsimonious model assumed that only the response threshold and drift rate had different group-level distributions. Inter-trial variability parameters were not included in our models due to the small number of trials available and to allow convergence.

**Model fit**. We followed the Bayesian Hierarchical Drift Diffusion Models[67] recommendations to fit our models. For each model, the starting values were set at the maximum a posteriori value to accelerate convergence. Bayesian inference was then performed by drawing 50,000 posterior samples by Markov Chain-Monte Carlo methods. The first 25,000 samples were discarded to limit the influence of starting values on posterior distributions. We retained every 10th sample to reduce autocorrelation within chains. We performed 20 runs of the same model, which were then combined to generate the final model. Parameter convergence was checked before analysis by visual examination of the trace, autocorrelation, and marginal posterior distribution, and with Gelman-Rubin R-hat statistic[69] comparing the within-chain and between-chain variances of the 20 different runs of the same model.

**Model selection**. To identify the model best fitting our data, we used the deviance information criterion[70], a measurement of goodness-of-fit for Bayesian hierarchical models with a penalty for the number of free parameters.

The difference in deviance information criterion between the full and parsimonious models was not significant (<10) (Supplementary Table 9), indicating that the two models fit equally well our set of data.

We also checked the ability of the models to generate the observed data by performing posterior predictive checks. We sampled 500 sets of parameters from the posterior distributions of the fitted models and simulated 500 sets of data corresponding to our original design (number of trials and participants). The posterior predictions were generated by averaging these 500 sets, and were compared with the observed data.

The posterior predictive checks showed that both models yielded data similar to the observed data. All reported statistics are in the 95% credible interval of the observed data (Supplementary Table 9 and Supplementary Fig. 3).

Finally, the parameters of the full and parsimonious models (e.g. correlations between the response thresholds of the two models) were all highly correlated (r(91) = 1, $p < 0.001$) for all parameters. Altogether, this indicates the lack of added value of the full model, hence our selection of the parsimonious model.

### Statistical analysis of structural imaging data

**MRI acquisition and preprocessing.** Three-dimensional, T1-weighted structural scans were acquired with a MP-RAGE sequence on a Siemens symphony 1.5 Tesla whole-body scanner (Henri Mondor Hospital, Paris, France) with a 12-channel head coil (TR = 2400 ms, TE = 3.72 ms, TI = 1000 ms, FA = 8°, FOV = 256*256 mm$^2$, 1 mm isotropic voxel, slice thickness = 1 mm, no inter-slice gap, 160 sagittal sections).

Cortical thickness is a sensitive method for studying cortical changes within the brain and is more sensitive to age or disease related grey matter changes than VBM[71,72] while being independent from confounding information such as surface area and cortical folding[73].

MRI scans were preprocessed with Freesurfer (http://surfer.nmr.mgh.harvard.edu/)[74]. The procedure included the removal of non-brain tissue, normalization of the intensity of the grey/white matter boundary, automated topology correction, and surface deformation. The following subcortical structures were automatically segmented: thalamus, striatum, pallidum, hippocampus, and amygdala. Cortical thickness (in mm) was calculated as the shortest distance between the grey/white matter boundary and the pial surface at each vertex across the cortical mantle[75]. All reconstructed data were visually checked for segmentation accuracy by a neuropsychologist (ML) trained in brain structural segmentation analysis, and reviewed by an expert neurologist blinded to participants' genetic makeup. The spherical cortical thickness data of all participants were mapped onto an "average" subject by surface-based registration methods[76] to morphologically match homologous cortical locations in participants. We used a 10 mm full width at half-maximum Gaussian kernel to smooth maps of cortical thickness.

**Neuroanatomical differences between groups.** The subcortical structures were compared between groups using a mixed ANOVA with subcortical volumes normalized to the total intracranial volume as the dependent variable, group as a between factor, subcortical structure as a within factor, and age as a covariate. We corrected post-hoc analysis for multiple comparisons with the Tukey method.

Vertex-wise comparisons of cortical thickness values between groups were performed on Freesurfer using generalized linear models, with cortical thickness as the dependent variable, group as the predictive factor, and age as a covariate. At each vertex, $F$-statistics were calculated to test the hypothesis of a difference in cortical thickness for each group comparison (two-tailed test). We corrected for multiple comparisons by family-wise error cluster-based correction, using Monte Carlo simulations with 10,000 iterations.

### Relationship between brain structure and DDM parameters in mutation carriers

For cortical analyses, we fitted a generalized linear model for each parameter with the cortical thickness as the dependent variable, the parameter and the disease stage as predictive variables, and age as a covariate. At each vertex, $F$-statistics were calculated to test the hypothesis of an interaction between cortical thickness and disease stage. We corrected for multiple comparisons by family-wise error cluster-based correction, using Monte Carlo simulations with 10,000 iterations. If there was no cluster with a significant interaction, the disease stage was removed from the analysis before testing the hypothesis of the non-null relationship (two-tailed test) between the DDMs parameter and the cortical thickness.

Finally, we used the clusters identified by the generalized linear model analyses as regions of interest from which we extracted cortical thickness values for imaging controls and mutation carriers. We tested for differences in cortical thickness between mutation carriers and imaging controls by performing ANOVA on the mean cortical thickness in each significant cluster.

### Statistics and reproducibility

Data handling, data representation and behavioural statistical analyses were done using the tidyverse toolbox 2.0.0[77] in R 4.2.3 within RStudio 2023.06.1[78]. We performed type 3 Anovas using the function *ezANOVA* from the ez package. Tukey post-hocs were done using the *TukeyHSD* function. Unless specified otherwise, *t*-tests were independent two-tailed *t*-tests with unequal variances and performed using the *t.test* function. Linear models were fitted using *lm* and Anova from *car* package. Post-hocs were performed using *emmeans* and *emtrends* functions.

DDM fit was done using the HDDM toolbox 0.6.0[67] under Python 2.6. MRI analyses were performed with Freesurfer 6.

### Reporting summary

Further information on research design is available in the Nature Portfolio Reporting Summary linked to this article.

## Data availability

All participants signed an informed consent form guaranteeing data confidentiality. The conditions of our ethics approval, including the ethical consent by participants, do not permit public archiving of anonymized study data. Readers seeking access to the data should contact Prof. Anne-Catherine Bachoud-Lévi. Access will be granted to named individuals in accordance with ethical procedures governing the reuse of sensitive data, including a research partnership and the completion of a data transfer agreement provided by the APHP. Legal copyright restrictions prevent public archiving of the UHDRS, which can be obtained from UHDRS® | — Huntington Study Group.

## Code availability

Analysis code is archived in a publicly accessible repository https://osf.io/894ae/?view_only=ec711857a221400384ef17e333c15e9f. It requires the following software: Freesurfer 6 (http://surfer.nmr.mgh.harvard.edu/)[74], the HDDM toolbox 0.6.0[67] under Python 2.6, and R 4.2.3.

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

## Acknowledgements

This work was supported by the *Agence Nationale de la Recherche* (**ANR-17-EURE-0017**, and **ANR-11-JSH2-006-1**). The work was funded by a grant from *Fondation Maladies Rares, programme Sciences Humaines et Sociales & Maladies Rares* awarded to Charlotte Jacquemot, a PhD grant from *Ecole Doctorale Sciences de la vie et de la santé* ED402 awarded to Lorna Le Stanc, and an interface contract (*Institut National de la Santé Et de la Recherche Médicale*—INSERM) awarded to Anne-Catherine Bachoud-Lévi. The Henri Mondor Hospital National Reference Centre for Huntington's Disease funded the follow-up of all patients included in this study (Ministry of Health). We thank Laurent Cleret de Langavant, Arnaud Cachia, and Sylvain Charron for their insight concerning the imaging methodology and results, Lucas Filipin for assistance with the software, and Renaud Massart for proofreading the manuscript.

## Author contributions

Conceptualization: C.J. and A.C.B.L. Data curation: L.L.S., M.L., A.C.B.L., and C.J. Data analysis and implementation of the computer code and algorithms: L.L.S. and M.L. Investigation: K.Y., A.S., M.G., A.C.B.L., and C.J. Methodology: L.L.S., M.G., A.C.B.L., and C.J. Project administration, supervision, and validation: A.C.B.L. and C.J. Resources, supplies, and patients recruitment: K.Y., A.C.B.L., and C.J. Software and computer programmes: L.L.S., M.L., and M.G. Visualization and data presentation: L.L.S. Drafting the first manuscript: L.L.S., M.L., and C.J. Rewriting—reviewing and editing the following versions of the manuscript: L.L.S., A.C.B.L., M.G., and C.J. Funding acquisition: C.J. and A.C.B.L.

## Competing interests
The authors declare no competing interests.
