## [Peer Review File · Communications Biology]

Reviewers' comments:

Reviewer #1 (Remarks to the Author):

The aim of the study is to assess the cognitive processes underlying compensation in premanifest Huntington's disease (preHD) (N=20) and manifest HD (N=28) in comparison to controls (N=45). Authors tried to disentangle cognitive dysfunction from compensation by using Drift diffusion models (DDM) in a language discrimination task developed by the authors, where participants are required to choose between two alternatives. Authors also divided preHDs into groups according to the time remaining until the predicted age-at-onset. Neuroanatomical data was used to explore the correlation between cognitive compensation mechanisms and brain structural adaptation to compensation mechanisms. Behavioural and neuroanatomical analyses showed the presence of compensation in preHD, and they behaved similarly to controls despite the presence of striatal atrophy. Specifically, DDMs revealed that compensation relies on decision making, with the higher drift rate in preHDs being a compensatory mechanism that preserves normal accuracy and response times; these findings also correlated with left superior parietal hypertrophy and hippocampal hypertrophy, suggesting that compensatory mechanisms could be related to attentional network.

The present manuscript is interesting and well written. The current literature is very poor respective to studying cognitive reserve in Huntington's disease, and to date there is no consensus on which are the best cognitive tools to assess that.

However, I have some major concerns prior to publication:

1. Authors refer to “cognitive reserve” and “cognitive compensation” interchangeably throughout the entire manuscript, and it is not clear which is the outcome of interest (the title reports “cognitive reserve”, but sometimes results refer to compensation instead). Could you please clarify their differences?
2. Authors decided to study compensation by analysing cognitive mechanisms in combination with a neuroanatomical approach (subcortical structures volumes and cortical thickness analyses). However, I suspect that this methodological approach does not fully capture and take into account the dynamic nature of compensation as a process. Previous studies (please refer to Kloppel et al., eBioMedicine 2015, Gregory et al., Brain 2018 for a few examples) used brain volumes as a composite score for “disease load”, but then used fMRI data (seed-region based correlation and effective connectivity) to study compensatory processes in the brain. In the absence of fMRI data, please include a section in the Discussion reporting these limitations in your study.
3. Introduction: it is not clear from the text the rationale and the added value of using DDMs to separate compensation from cognitive dysfunction in comparison to other approaches. Could you please provide further details on this aspect? Furthermore, has the Language discrimination task been validated and used as a reliable measure to determine cognitive reserve (also in the Huntington's disease field)?

4. Could authors provide any data relative to premorbid intelligence of their participants? These scores could be used as additional covariates for their analyses on cognitive compensation.

5. Imaging controls: authors state that these participants “did not perform the language task and the cognitive battery”. How did authors ensure that these participants were control participants?

6. Results: authors split preHDs into three groups according to their time to predicted age-at-onset; however, the number of participants for each category (N=4, 7 and 9 respectively) is very small to infer any robust and significant findings, and these results should be interpreted with caution. Please include this as a limitation of your study.

Reviewer #2 (Remarks to the Author):

The paper uses a combination of computational model of decision making and neuroimaging evidence to test for the presence of cognitive reserve in a genetical model of a neurodegenerative disease, the Huntington disease (HD). In the case of neurodegenerative disease, observing cognitive reserve is complicated by the fact that the time of diagnosis is often already associated with cognitive decline. By studying a genetic model of neurodegenerative disease like the HD, the authors are able to evaluate the role of cognitive reserve prior to manifest cognitive deficits. The sample of patients and controls for the study proves to be well chosen based on the presence of a difference between earlyHD and the controls, and absence of evidence towards a difference between preHD and controls. On the other hand the presence of brain anatomy as a difference between preHD and controls, despite no evidence for behavioral differences, is also a very interesting context to look for cognitive reserve. On the research question itself, the higher drift rate as a compensatory mechanism to counteract the ongoing behavioral performance degradation is a major addition to the understanding of neurodegenerative diseases, the time-course of those diseases, and the role of cognitive reserve. The drift rate and more generally computational models of decision making as a measurable cognitive marker of compensation is indeed an interesting suggestion for future research. The discussion is compelling and very nicely links previous findings on both healthy and HD patients to the findings of the present paper. It also clearly states the limits of the current study. Overall, in addition to the better comprehension it offers on the role of cognitive reserve in neurodegenerative diseases, I see the manuscript as a very interesting application of computational models of behavior, here decision-making, to a clinical context.

Now several major points limit the degree to which I see the conclusions of the authors as supported by the data. As a disclaimer, some of these major points are derived from the ideal experimental case scenario where the recruitment of participants is easier than in the case of HD patients.

Major:

On the structural MRI:

- The authors don't mention any correction for multiple testing for the post-hoc tests, which is

necessary in this case as they perform at least 15 statistical tests. I do believe however that the main result from that section (preHD striatal atrophy) wouldn't change given an appropriate correction, hence this finding is probably not challenged.

- In general, as many other papers, authors conclude for absence for evidence when there is really only no evidence for presence (e.g. structural analysis "there was no difference between groups concerning the volumes of the hippocampus and amygdala (all $p > .05$)" or e.g. combined MRI-DDM section "In another word, the observed changes in brain anatomy were specifically related to the increase in drift rate.") which is frequent error but should still be corrected in the manuscript. In this section, I believe that the cortical thinning analysis statistics should be reported in the main text instead of the appendix if the authors choose to give the conclusions in the main text.

On the DDM section:

- Ideally the reference to the DDM should point to the specific formulation of the DDM that was used by the authors (e.g. Ratcliff, 1978 for the "simple" DDM or Ratcliff & Tuerlinckx, 2002, if including drift rate/non-decision/starting point variabilities), this would allow readers to understand what model was actually implemented.

- More importantly, the 2-step statistical procedure (1st step - model parameter estimation and 2nd step - statistical analysis on resulting parameters) is relatively common but suboptimal in terms of hypothesis testing (see Boehm, Marsman, Matzke, and Wagenmakers, 2018). Given that authors already use HDDM to fit the models, it wouldn't be too complicated to have a one-step approach to model fitting (e.g. fitting linear models on all parameters with an intercept and a categorical predictor for group), see <https://hddm.readthedocs.io/en/latest/howto.html#stimulus-coding-with-hddmregression>. Based on the reading of the supplementary material, I understand that this might already be implemented at least for the group comparison on drift and boundaries, therefore the authors could simply report the posterior distribution for the population mean on these two parameters. For the following section, amongst others the relationship between DDM parameter and MRI should also be done this way in order to have correct inferences on these relations. In addition to the inappropriateness of this two-stage testing, it is not a good practice to ignore all uncertainty in parameter estimates and reduce them to point-estimate as is being done here. The previously described regression would also alleviate that problem. I am not certain whether all the evidence reported in this section will remain the same after taking this step towards a better modelisation.

- While very interesting for investigating the cognitive reserve, the analysis splitting preHD into three groups is not very convincing as the repartition seems arbitrary (why 3 groups? why 10 and 1 years?). In the absence of pre-registration or strong theoretical justifications of group splitting, it would be more convincing to actually model this bell-shape and provide evidence (or not) in favor of that shape (e.g. polynomial regression with age-to-onset as predictor). This also applies to the next section.

- On the link between drift rate and response threshold I fail to understand how this is related to the authors' research question. Overall I think this paragraph lacks clarity for the reader to understand.

- Finally the method section and the corresponding supplementary material section don't allow readers to understand what type of model was really fitted. Were inter-trial variability parameters included in the model? If so, estimated for each individual, group or for the whole sample?

- Also providing a visual summary of the model fits for the different groups would allow the reader to

have a critical assessment of the appropriateness of the fitted model. In the current report it is impossible to appreciate the goodness of fit of the model and whether it differs between participants

On the combined neuro-anatomical and DDM section:

- Overall this section is hard to follow, a more detailed header would allow reader to follow along more easily
- The linear model parameters (not only the t values) should be reported in order to understand the strength of the relationship between drift rate and cortical thickness. Particularly in the second paragraph of the section.

Minor:

- p.6 2nd paragraph, what does “least” next to the p-value mean?
- p.7 “On the contrary” statistics are missing in the main text for this paragraph
- p.9 first paragraph, Brain reserve should be introduced in the introduction as this seems to be an interesting and important concept in this research.
- The description of the DDM fitting strategy should be in the method section rather than in the supplementary material.
- Appendix
- Supplementary Table 1: the leading zeros makes this table hard to read, maybe use a log base 10 transformation to make it more easily readable
- Supplementary Figure 1. hard to see, maybe put the figure row-wise instead of column-wise to be able to better see the areas presenting cortical thickness differences
- Model selection refers to Table 2 while it should probably be Table 5

Gabriel Weindel

Reviewer #3 (Remarks to the Author):

This study collected behavioural data from a binary auditory discrimination task from earlyHDs, preHDs and controls. EarlyHDs had lower accuracy and slower RT than the rest two groups, and preHDs had comparable performance as controls. The authors fitted DDMs to the behavioural data. Group differences in behavioural performance yielded lower decision thresholds in earlyHDs (compared with preHDs and controls). On drift rate, it is the lowest in the earlyHDs. Further subgroup analysis indicated that among preHDs, the drift rates differed according to their time to predicted age-at-onset, for which the authors interpreted as a compensatory mechanism. The manuscript is in general easy to follow. However, the authors need to consider the following issues in their analyses and results.

1. Cognitive compensation. The manuscript reported no significant difference in behavioural measures between preHDs and controls. Throughout the manuscript (Intro, results and discussion), the lack of behavioural difference together with fitted DDM parameters were used to support a compensation hypothesis: drift rate was increased to compensate for the decrease in

decision threshold (e.g., line 301). However, I feel this proposition does not offer new insights (in terms of cognitive compensation) beyond its immediate interpretation of the data, and, in fact, the data cannot be adequately summarised by this statement. First, a larger decision threshold in earlyHDs vs controls does not necessarily imply a decision deficit but can simply be a more conservative trade-off towards decision accuracy over speed in earlyHDs. Second, relating to this, the increase in HD's decision threshold in HDs does not mean more impulsive decisions, opposite to the claim in the manuscript. Third, there is no group difference in the threshold between preHDs and controls, and the manuscript did not report any bell shaped parameter values from different preHD subgroups. Together, I felt that the current study does report some interesting findings of structural correlates of the drift rate, but the results do not provide clear support for a compensatory mechanism.

2. Models and statistics. First, the model assumed the same absolute drift rate value between "same" and "different" choices. Please provide evidence to support this model design decision. For example, could the author confirm that there was no difference in any behavioural measures between the two choices (e.g., Bayesian statistics supporting the null is appropriate here)? Second, the manuscript opted for the "parsimonious" model over the "full" model. Considering that the two models have very close DIC values, could the authors confirm that their main conclusions are not changed if using parameters from the full model? Third (please correct me if I am wrong), throughout the manuscript, point estimates of individual subject's model parameters were used for frequentist statistics (e.g., ANOVA and regression). I believe such calculations are not valid, because individuals' posterior model parameters are sampled from group-level posteriors. As a result, they are no longer independent of each other and hence are not fit for subsequent frequentist statistics. A more proper method is to do a Bayesian comparison directly on the group-level posteriors. For regression, the authors should consider using regression models during MCMC (i.e., adding MRI-derived measures as regressors during HDDM model fit).

3. Model fit. Supplementary Table 5. Please report posterior model checks for each group. In addition, the authors should clarify the use of "Proportion of "same" responses" as a posterior check metric. (1) It appears that the fitted value has very large std; and (2) It is unclear how it relates to the high accuracies observed in behavioural data.

4. PreHD subgrouping. Please provide a rationale for the subgrouping criteria used in the study. Why were <1 yr and >10 yrs used to split the group into three unequal subgroups? With N as small as =4 in one subgroup, are the authors confident that their results are not strongly susceptible to type II errors?

5. Cortical thickness analysis. I am not sure if this analysis is circular and I hope the authors can clarify. The cortical cluster was first identified as a significant correlation with the drift rate across all subjects (line 246), and the cortical thickness from the cluster was then compared between groups and preHD subgroups. Since we already know that the drift rate differs between groups, the subsequent ANOVA and t-tests are not independent of the whole-brain analysis.

Minor,

1. I am puzzled to understand why the behavioural task is termed a language task since only pseudo-words were used.

2. Line 337 Fig. 3F  Fig. 3E?

3. On several occasions (e.g., line 238-239), $t(33)=xx$ was reported for different groups or preHD subgroups. Please check the DoF as they should not be all the same (given the different numbers of subjects in each group)?

Point-by-point response to reviewers

Reviewer #1 (Remarks to the Author):

The aim of the study is to assess the cognitive processes underlying compensation in premanifest Huntington's disease (preHD) (N=20) and manifest HD (N=28) in comparison to controls (N=45). Authors tried to disentangle cognitive dysfunction from compensation by using Drift diffusion models (DDM) in a language discrimination task developed by the authors, where participants are required to choose between two alternatives. Authors also divided preHDs into groups according to the time remaining until the predicted age-at-onset. Neuroanatomical data was used to explore the correlation between cognitive compensation mechanisms and brain structural adaptation to compensation mechanisms. Behavioural and neuroanatomical analyses showed the presence of compensation in preHD, and they behaved similarly to controls despite the presence of striatal atrophy. Specifically, DDMs revealed that compensation relies on decision making, with the higher drift rate in preHDs being a compensatory mechanism that preserves normal accuracy and response times; these findings also correlated with left superior parietal hypertrophy and hippocampal hypertrophy, suggesting that compensatory mechanisms could be related to attentional network.

The present manuscript is interesting and well written. The current literature is very poor respective to studying cognitive reserve in Huntington's disease, and to date there is no consensus on which are the best cognitive tools to assess that.

However, I have some major concerns prior to publication:

1. Authors refer to "cognitive reserve" and "cognitive compensation" interchangeably throughout the entire manuscript, and it is not clear which is the outcome of interest (the title reports "cognitive reserve", but sometimes results refer to compensation instead). Could you please clarify their differences?

We would like to thank the reviewer for pointing out this ambiguous term and his comment helped us to clarify this conceptual point throughout the paper. The concept of "reserve" refers to the capacity of the brain to resist neuropathological changes and preserve cognitive functioning. Reserve is thought to rely on brain reserve and cognitive reserve. While brain reserve relies purely on quantitative aspects of the brain such as brain size for example, cognitive reserve refers to differences in cognitive functions that explain the discrepancy between the neuropathological load and the behavioral outcome. Cognitive reserve relies on two concepts: neural reserve and neural compensation. Neural compensation refers to the ability of the brain to use alternative networks not typically used in healthy individuals to maintain cognitive performances while neural reserve refers to the ability to increase network efficiency and or capacity of the network typically used in unimpaired individuals to achieve the task. For a review of the different concepts see Barulli, D., and Stern, Y. (2013).

Throughout our manuscript we are discussing cognitive reserve and control for the brain reserve using total intracranial volume as a proxy when looking at the brain substrate of the reserve. We used cognitive compensation as a term to try to convey the idea that there is a cognitive reorganization to compensate for the neuropathological load. We realize that this can be confusing given the terms defined in the literature and we now only refer to cognitive reserve.

We have rewritten the abstract and the first paragraph of the introduction to better explain these different concepts (cognitive reserve/neural reserve/compensation) (l. 44-53).

*Neurodegenerative diseases affect brain parts and functions at variable degrees and at different stages over the course of the disease, and eventually precipitate brain atrophy that precedes intellectual deterioration*¹. In general, patient's normal behavior is maintained until the

neuropathological damage surpasses the adaptive capabilities of the brain leading to the appearance of clinical symptoms²⁻⁴. The concept of “reserve” refers to this capacity of the brain to resist neuropathological changes and preserve cognitive functioning. Reserve is thought to rely on brain reserve and cognitive reserve. While brain reserve relies purely on quantitative aspects of the brain such as brain size for example, cognitive reserve reflects the brain’s capabilities to optimize and develop alternative cognitive strategies to actively preserve cognitive functions. Cognitive reserve depends on patient’s lifetime intellectual activities and environmental factors. It relies on two concepts: neural reserve and neural compensation. The brain can either increase the efficiency of an existing yet deteriorating network (neural reserve) and/or recruit other regions upon performing a task (neural compensation)^{3,5}; i.e., some cognitive functions may compensate for others that were impacted at earlier stages.

2. Authors decided to study compensation by analysing cognitive mechanisms in combination with a neuroanatomical approach (subcortical structures volumes and cortical thickness analyses). However, I suspect that this methodological approach does not fully capture and take into account the dynamic nature of compensation as a process. Previous studies (please refer to Kloppel et al., eBioMedicine 2015, Gregory et al., Brain 2018 for a few examples) used brain volumes as a composite score for “disease load”, but then used fMRI data (seed-region based correlation and effective connectivity) to study compensatory processes in the brain. In the absence of fMRI data, please include a section in the Discussion reporting these limitations in your study.

We thank the reviewer for pointing out that this point was not clear enough in the previous version. We have added the references and expanded the paragraph that discusses the limitations of the study (l. 423-426 and l. 441-446).

Overall, whether the compensatory mechanism is induced by attention modulation which refers to neural reserve, or attention recruitment which refers to neural compensation, is not resolved here. The fact that preHDs did not show cortical atrophy of the attentional system (see supplementary Fig.1, Supplementary Table 2) indicates that this network was not a priori affected. Future research replicating the results using functional MRI data and tasks aimed at testing attentional capacities in HD mutation carriers would allow to confirm the overactivation or additional recruitment of the attentional network and to capture the dynamic nature of cognitive reserve. It is worth mentioning here that our imaging results corresponded to cognitive reserve rather than to brain reserve since we took total intracranial volume into account in cortical thickness analyses and normalized volumes in subcortical analyses.

Despite the study small cohort size, DDMs detected differences between preHDs and controls and identified cognitive processes that might underlie compensatory mechanisms in a discrimination task. We have shown that DDMs, combined with a language task, had an added value over classical neuropsychological tests, which rarely detect differences in preHDs, unless hundreds of participants are recruited^{1,59}. Choosing a language task was motivated by studies showing the sensitivity of language in detecting subtle disorders in small cohorts of preHDs¹⁴⁻¹⁸. The duration of the task and its simplicity (pseudoword discrimination) made it easily adaptable and transferable to other languages and other brain pathologies. Showing the generality of the drift rate as a marker of compensation in preHDs, i.e., of the increase in attention allocation as a compensatory mechanism, would require assessing other cognitive domains. Structural imaging provides a hint into the attentional network. Yet, it does not fully capture and take into account the dynamic nature of cognitive reserve as a process and does not allow discriminating between neural reserve or neural compensation for example.

Studying the functional correlates of the drift rate in Huntington's disease would provide a greater understanding of online allocation of attentional resources as a compensatory mechanism^{4,60}. These findings suggest that focusing on the role of the superior parietal cortex and hippocampus is promising in Huntington's disease, albeit needs to be replicated in other neurodegenerative diseases. Strengthening the cognitive mechanisms that compensate for the degeneration of the brain in neurodegenerative diseases might allow to delay the onset and progression of these diseases and offer a promising line of treatment for neurodegenerative diseases.

3. Introduction: it is not clear from the text the rationale and the added value of using DDMs to separate compensation from cognitive dysfunction in comparison to other approaches. Could you please provide further details on this aspect? Furthermore, has the Language discrimination task been validated and used as a reliable measure to determine cognitive reserve (also in the Huntington's disease field)?

We have rephrased the paragraph to better explain the rationale for choosing MDDs to study cognitive reserve in participants with Huntington's disease who have brain atrophy but no clinical signs of the disease. We also explain the reasoning behind the choice of the language discrimination task (in Huntington's disease) combined with the DDMs.

The third paragraph of the introduction now reads as follows (l. 70-84):

To study the mechanisms of cognitive reserve separately from cognitive dysfunction¹⁰⁻¹³, we selected a cognitive task that is impaired in the early stages of the disease (no longer effective cognitive reserve) but still normal in presymptomatic subjects (effective cognitive reserve). We used a language discrimination task as language is one of the first cognitive function to decline in Huntington Disease¹³⁻¹⁸, with normal or near-normal performance in preHDs and abnormal performance in earlyHDs, suggesting that language performance would be a reliable measure to study cognitive reserve. We analyzed the results of the language task - deciding whether two items are similar or not - through Drift diffusion models (DDMs)^{19,20}. These models enable us to evaluate separately the cognitive parameters involved in the discrimination task: accumulation over time of sensory evidence at a certain rate up to a response threshold that triggers the motor response to indicate which of the two alternatives to pick (similar or not). The hypothesis is that in preHD, one of the impaired cognitive parameters will be compensated for by another cognitive parameter, resulting in normal behavioral performance, whereas in earlyHD, the compensatory cognitive parameter will no longer be effective, resulting in a behavioral deficit.

4. Could authors provide any data relative to premorbid intelligence of their participants? These scores could be used as additional covariates for their analyses on cognitive compensation.

Unfortunately, the IQ is not measured in our protocols. Nevertheless, the three groups are matched for educational level (see Table 1, and Methods, participants section). We chose to match groups rather than to include additional covariates in the statistical analyses in order to avoid running underpowered statistical analyses. In addition, to make sure that this consideration would not affect the study of cognitive reserve, we chose a low-level task, i.e. a phonological discrimination task with two items to discriminate. This linguistic task is based on the phonological process, which is automatic and acquired very early in infancy (Dehaene-Lambertz and Baillet, 1998; Näätänen et al., 1997).

5. Imaging controls: authors state that these participants “did not perform the language task and the cognitive battery”. How did authors ensure that these participants were control participants?

The control participants were healthy subjects who had no previous or current neurological or psychiatric history. The anatomical MRI of these control participants were checked for any abnormalities by neuroradiologists and used in previous studies (Giavazzi, et al.,2018; Le Stanc, et al., 2023).

This information was added (l. 150-151):

They were compared with 30 scans from a cohort of external healthy participants who did not perform the language task and the cognitive battery (imaging controls). The imaging controls had no previous or current neurological or psychiatric history and their anatomical MRI was checked by a neuroradiologist for any abnormalities.

6. Results: authors split preHDs into three groups according to their time to predicted age-at-onset; however, the number of participants for each category (N=4, 7 and 9 respectively) is very small to infer any robust and significant findings, and these results should be interpreted with caution. Please include this as a limitation of your study.

We agree that the number of participants is small and we have taken care in the paper to interpret the results with caution; these results should be replicated in a larger cohort. We have added this limitation in the discussion (l 407-416).

Cognitive reserve that operates through active compensation mechanisms may depend on either an increase in the activity of a deficient network (neural reserve) or the recruitment of alternative networks with available resources (neural compensation)⁵. Both have been observed in preHDs where functional imaging studies showed changes in BOLD responses in task-dependent regions despite similar behavioral performances to that of controls^{4,30,31}. Changes in connectivity, structure, or activation can provide information about the link between the disease and neural reorganization. However, this link might be pathological rather than compensatory if it is not correlated with improved performances^{2,3}. Here, we observed left superior parietal cortex hypertrophy in preHDs (Fig. 4D). Such an increase in cortical thickness, supposedly caused by hyperactivation, was associated with better performances (shorter response times, better accuracy, and higher drift rates) (Supplementary Fig. 2, Supplementary Table 4, Fig. 4D,E), supporting the hypothesis of a successful compensation, as previously reported in motor learning²⁹. The post-hoc analysis showed that only preHDs middle group had this cortical hypertrophy, not preHDs far from and close to predicted age-at-onset; a bell-shaped pattern in favor of a compensatory mechanism emerging at a certain point and failing as the pathological load increases. Although these findings would need to be replicated in a larger cohort, they are consistent with the results of previous studies on Alzheimer’s disease⁵³ and Huntington’s disease⁵⁴ which reported a preclinical stage of hypertrophy preceding atrophy in symptomatic patients. This may reflect an experience-dependent increase in neural volume^{55,56} as an attempt to compensate for the dysregulation of the striatal network.

Reviewer #2 (Remarks to the Author):

The paper uses a combination of computational model of decision making and neuroimaging evidence to test for the presence of cognitive reserve in a genetical model of a neurodegenerative disease, the Huntington disease (HD). In the case of neurodegenerative disease, observing cognitive reserve is

complicated by the fact that the time of diagnosis is often already associated with cognitive decline. By studying a genetic model of neurodegenerative disease like the HD, the authors are able to evaluate the role of cognitive reserve prior to manifest cognitive deficits. The sample of patients and controls for the study proves to be well chosen based on the presence of a difference between earlyHD and the controls, and absence of evidence towards a difference between preHD and controls. On the other hand the presence of brain anatomy as a difference between preHD and controls, despite no evidence for behavioral differences, is also a very interesting context to look for cognitive reserve. On the research question itself, the higher drift rate as a compensatory mechanism to counteract the ongoing behavioral performance degradation is a major addition to the understanding of neurodegenerative diseases, the time-course of those diseases, and the role of cognitive reserve. The drift rate and more generally computational models of decision making as a measurable cognitive marker of compensation is indeed an interesting suggestion for future research. The discussion is compelling and very nicely links previous findings on both healthy and HD patients to the findings of the present paper. It also clearly states the limits of the current study. Overall, in addition to the better comprehension it offers on the role of cognitive reserve in neurodegenerative diseases, I see the manuscript as a very interesting application of computational models of behavior, here decision-making, to a clinical context.

Now several major points limit the degree to which I see the conclusions of the authors as supported by the data. As a disclaimer, some of these major points are derived from the ideal experimental case scenario where the recruitment of participants is easier than in the case of HD patients.

Major:

On the structural MRI:

- The authors don't mention any correction for multiple testing for the post-hoc tests, which is necessary in this case as they perform at least 15 statistical tests. I do believe however that the main result from that section (preHD striatal atrophy) wouldn't change given an appropriate correction, hence this finding is probably not challenged.

We thank the reviewer for pointing out that this information was missing in the paper. We added this information in Methods section (l. 656-660) and Table 1 of the Supplementary Results. This information is also reported in the Results section (l. 158). Post-hoc were realized with the Tukey method for each sub-cortical structure as implemented in (<https://cran.r-project.org/web/packages/emmeans/>). The p-values were adjusted with the Tukey method for comparing a family of 3 estimates corresponding to the three comparisons (earlyHDS vs imaging controls, earlyHDs vs preHDs and preHDs vs imaging controls). Correcting for the 15 statistical tests appears to us a little too strict. That being said, we performed a Bonferroni correction to make sure that our results stand. As predicted by the reviewer, all significant results remain significant.

Results section

We first compared the volumes of subcortical structures (striatum, pallidum, thalamus, hippocampus, and amygdala) between the mutation carriers and imaging controls. We ran mixed ANOVA with subcortical volumes normalized to the total intracranial volume as the dependent variable, group as a between factor, subcortical structure as a within factor, and age as a covariate. There was an interaction between subcortical structure and group ($F(8,372) = 34.32, p < .001$). Tukey's post-hoc showed that earlyHDs had lower grey matter volumes than both imaging controls and preHDs, in the striatum (all $p < .001$), pallidum ($p = .0510$, least), and thalamus (all $p < .01$) (Fig. 2, Supplementary Table 1).

Methods section

The subcortical structures were compared between groups using a mixed ANOVA with subcortical volumes normalized to the total intracranial volume as the dependent variable, group as a between factor, subcortical structure as a within factor, and age as a covariate. We corrected post-hoc analysis for multiple comparisons with the Tukey method.

Legend of supp Table 1

Mean subcortical volumes. Descriptive statistics (mean±standard deviation) of each group. Group comparisons: Tukey's post-hoc test [95% confidence interval] and p-value of each pair comparison corrected for the three multiple comparisons. preHDs: premanifest participants; earlyHDs: early-stage Huntington's disease patients.

- In general, as many other papers, authors conclude for absence for evidence when there is really only no evidence for presence (e.g. structural analysis "there was no difference between groups concerning the volumes of the hippocampus and amygdala (all $p > .05$)" or e.g. combined MRI-DDM section "In another word, the observed changes in brain anatomy were specifically related to the increase in drift rate.") which is frequent error but should still be corrected in the manuscript.

We thank the reviewer for this remark and modified the manuscript as follow :

L. 137: On the other hand, the preHDs' performances were statistically similar to those of the controls (all $p > .05$) (Table 2).

L. 141: As for the mean accuracy and mean response time of the language task, analyses showed a significant main effect of group (accuracy: $F(2,90) = 20.93, p < .001$); response time: $F(2,90) = 41.06, p < .001$). Tukey's post-hoc analyses revealed that the earlyHDs were less accurate and slower than controls and preHDs. In contrast, the preHDs were not statistically different from the controls (Table 3).

L. 162: Overall, there was no statistical evidence for a difference between groups concerning the volumes of the hippocampus and amygdala (all $p > .05$).

L. 277: There were no other significant differences between any of the sub-groups of preHDs and imaging controls (Fig. 4H).

L. 286: In another word, there is no statistical evidence that the observed changes in brain anatomy were not specifically related to the increase in drift rate.

In this section, I believe that the cortical thinning analysis statistics should be reported in the main text instead of the appendix if the authors choose to give the conclusions in the main text.

We thank the reviewer for this suggestion and have now reported the cortical thinning analysis statistics in the main text, in the results section (l. 164-169).

The following step was to study the differences in cortical thickness. EarlyHDs showed cortical thinning in the left angular gyrus, the left occipital superior cortex, the right caudal part of the middle frontal cortex, and the right lateral occipital lobe upon comparing them with imaging controls (supplementary Fig. 1A and Table 2). EarlyHDs also had a thinner right lateral occipital cortex than that of preHDs (supplementary Fig. 1B and Table 2). Cortical thickness was similar in preHDs and imaging controls.

On the DDM section:

- Ideally the reference to the DDM should point to the specific formulation of the DDM that was used by the authors (e.g. Ratcliff, 1978 for the “simple” DDM or Ratcliff & Tuerlinckx, 2002, if including drift rate/non-decision/starting point variabilities), this would allow readers to understand what model was actually implemented.

As suggested by the reviewer and in order to clarify the model implemented, we change the reference to Ratcliff 1978. Of note, we could not implement the more complex model due to the reduced quantity of trials that we have.

- More importantly, the 2-step statistical procedure (1st step - model parameter estimation and 2nd step - statistical analysis on resulting parameters) is relatively common but suboptimal in terms of hypothesis testing (see Boehm, Marsman, Matzke, and Wagenmakers, 2018). Given that authors already use HDDM to fit the models, it wouldn't be too complicated to have a one-step approach to model fitting (e.g. fitting linear models on all parameters with an intercept and a categorical predictor for group), see <https://hddm.readthedocs.io/en/latest/howto.html#stimulus-coding-with-hddmregression>. Based on the reading of the supplementary material, I understand that this might already be implemented at least for the group comparison on drift and boundaries, therefore the authors could simply report the posterior distribution for the population mean on these two parameters. For the following section, amongst others the relationship between DDM parameter and MRI should also be done this way in order to have correct inferences on these relations. In addition to the inappropriateness of this two-stage testing, it is not a good practice to ignore all uncertainty in parameter estimates and reduce them to point-estimate as is being done here. The previously described regression would also alleviate that problem. I am not certain whether all the evidence reported in this section will remain the same after taking this step towards a better modelisation.

We thank the reviewer for pointing to the HDDM regression method. This analysis was first performed but due to the lack of data, the model failed to converge. We therefore decided to turn to the current method.

Concerning the posterior distributions, the post-hoc division of the preHDs into three small groups does not allow running a DDM for each sub-group. Due to the number of trials and participants, such models fail to converge. Reporting posterior distribution for these groups is therefore impossible as well as performing inferences based on these posterior distributions. For the purpose of clarity in the paper we avoided reporting different types of statistics and for a graphical coherence between the different figures in the manuscript, we decided to report the results as boxplots in figure 3 rather than posterior distributions.

Running regression with the MRI data at once is technically impossible as the MRI data for the imaging control group and the DDM data from the behavioral control group do not come from the same participants.

- While very interesting for investigating the cognitive reserve, the analysis splitting preHD into three groups is not very convincing as the repartition seems arbitrary (why 3 groups? why 10 and 1 years?). In the absence of pre-registration or strong theoretical justifications of group splitting, it would be more convincing to actually model this bell-shape and provide evidence (or not) in favor of that shape (e.g. polynomial regression with age-to-onset as predictor). This also applies to the next section.

We thank the reviewer for pointing out that the group splitting was not sufficiently justified. We have added references in the new version of the article (l. 214-220) and acknowledged in the discussion section that the small number of subjects in each group is a limitation of our study (l. 412-416).

To further investigate this pattern, we split the preHDs into three groups according to their time to predicted age-at-onset ⁷. Determination of disease onset in Huntington's disease is made by clinical experience, but the conversion is a progressive process which makes it difficult to determine the exact moment of motor onset of HD ²⁴. PreHD were thus stratified into three groups, far, middle and close to the onset of the disease, according to the time remaining until the predicted onset of the disease: close to onset with predicted onset within a year (N=4), far to disease onset with a predicted onset ≥ 10 years (N=7) and middle onset with a predicted onset between 1 to 10 years (N=9) ²⁵.

Although these findings would need to be replicated in a larger cohort, they are consistent with the results of previous studies on Alzheimer's disease ⁵³ and Huntington's disease ⁵⁴ which reported a preclinical stage of hypertrophy and increased functional connectivity between the left caudate nucleus and parietal lobe preceding atrophy in symptomatic patients ³².

As for the suggestion of using polynomial regression, this would only have been possible if the time remaining until the predicted onset of the disease could be estimated for all subjects. This is not the case, since it is only possible for preHDs so it does not allow to compare the progression to controls and earlyHDs. For preHDs, we tried to run both a linear and a quadratic regression but none of the model fitted the data properly, probably due to the small sample size of the preHDs (linear model : $F(2,17) = 0.9139$, p-value: 0.4198 ; quadratic model = $F(1,18) = 0.4784$, p-value = 0.498) and there was no evidence that one model was best fitting the data than the other when comparing the two models ($F(1, 18) = 1.3403$, p-value = 0.263).

- On the link between drift rate and response threshold I fail to understand how this is related to the authors' research question. Overall I think this paragraph lacks clarity for the reader to understand.

We thank the reviewer for pointing out the section was not clear enough. We have rephrased it (l. 228-232 & 236-237).

The participant's behavioural performance is the result of the drift rate and response threshold parameters. Drift rate appears to show a compensatory pattern (faster rate of evidence accumulation) while response threshold shows a deteriorative pattern (higher response threshold) throughout all stages of the disease. We went further and investigated the relationship between drift rate and response threshold. We fitted a linear model with the drift rate as a dependent variable, response threshold and group as predictors, and age as a covariate. This showed that an increase in response threshold was associated with an increase in drift rate. The increase was similar in controls and earlyHDs, and sharper in preHDs (Table 4, Fig. 3E), meaning that for preHDs the drift rate might compensate for the increasing threshold allowing to preserve behavioral performances. This relationship between parameters was not induced by contamination between them as they were all not correlated (all $p > .05$) ²⁵.

- Finally the method section and the corresponding supplementary material section don't allow readers to understand what type of model was really fitted. Were inter-trial variability parameters included in the model? If so, estimated for each individual, group or for the whole sample?

We thank the reviewer for pointing out the lack of clarity of the manuscript. We added information in the method section to clarify and justify the type of model that was fitted. See l. 588-629.

Model fit and selection

Bayesian hierarchical DDM ⁶² is currently the most efficient method for dealing with a small number of observations ⁶³, hence its use in our work. It assumes that individual parameter estimates are random samples of group-level distributions. Data were cleaned as in behavioral analyses, albeit Bayesian

hierarchical DDMs used both correct and incorrect responses and response time. We assumed the same absolute drift rate value for both answers (“same” and “different”), a necessary hypothesis to estimate a possible relative bias toward one of the answers.

We tested two variants of the Bayesian hierarchical DDMs, full versus parsimonious. In the full model, each parameter had three group-level distributions, corresponding to our three groups (controls, preHDs, earlyHDs). The parsimonious model assumed that only the response threshold and drift rate had different group-level distributions. Inter-trial variability parameters were not included in our models due to the small number of trials available and to allow convergence.

Model fit

We followed the Bayesian Hierarchical Drift Diffusion Models (Wiecki et al., 2013) recommendations to fit our models. For each model, the starting values were set at the maximum a posteriori value to accelerate convergence. Bayesian inference was then performed by drawing 50,000 posterior samples by Markov Chain-Monte Carlo methods. The first 25,000 samples were discarded to limit the influence of starting values on posterior distributions. We retained every 10th sample to reduce autocorrelation within chains. We performed 20 runs of the same model, which were then combined to generate the final model. Parameter convergence was checked before analysis by visual examination of the trace, autocorrelation, and marginal posterior distribution, and with Gelman-Rubin R-hat statistic (Gelman and Rubin, 1992) comparing the within-chain and between-chain variances of the 20 different runs of the same model.

Model selection

To identify the model best fitting our data, we used the deviance information criterion (Spiegelhalter et al., 2002), a measurement of goodness-of-fit for Bayesian hierarchical models with a penalty for the number of free parameters.

The difference in deviance information criterion between the full and parsimonious models was not significant (<10) (Supplementary Table 5), indicating that the two models fit equally well our set of data. Following the recommended procedures to fit and assess model convergence⁶² (Supplementary Methods 1), we selected the parsimonious model since the full one did not capture any additional data patterns (Supplementary Methods 2, Supplementary Table 5 and Supplementary Figure 3).

We also checked the ability of the models to generate the observed data by performing posterior predictive checks. We sampled 500 sets of parameters from the posterior distributions of the fitted models and simulated 500 sets of data corresponding to our original design (number of trials and participants). The posterior predictions were generated by averaging these 500 sets, and were compared with the observed data.

The posterior predictive checks showed that both models yielded data similar to the observed data. All reported statistics are in the 95% credible interval of the observed data.

Finally, the parameters of the full and parsimonious models (e.g. correlations between the response thresholds of the two models) were all highly correlated ($r(91)=1$, $p<0.001$) for all parameters). Altogether, this indicates the lack of added value of the full model, hence our selection of the parsimonious model.

- Also providing a visual summary of the model fits for the different groups would allow the reader to have a critical assessment of the appropriateness of the fitted model. In the current report it is impossible to appreciate the goodness of fit of the model and whether it differs between participants

Supplementary Table 5 was intended to provide the information about the goodness of fit but we added a graphical representation of the information by plotting the posterior predictive checks for the different groups and the two different models in Supplementary Figure 3 and added the reference to it in the main manuscript. We hope that it helps clarify the appropriateness of fit.

Supplementary Figure 3. Posterior predictive checks for the full and parsimonious models The distribution along the positive x-axis shows the reaction time distribution for “different” trials in green while the distribution along the negative x-axis shows the reaction time distribution for “same” trials in yellow. Each panel shows the normalized histograms of the observed data and the model prediction (solid black line) for each group (controls, preHDs and earlyHDs).

On the combined neuro-anatomical and DDM section:

- Overall this section is hard to follow, a more detailed header would allow reader to follow along more easily

We thank the reviewer for this suggestion and have added more detailed headers for the different paragraphs of “Cognitive reserve is correlated with brain structures hippocampus volume and left superior parietal cortical thickness in mutation carriers:

L. 253, Drift rate is associated with hippocampus volume in preHDs

L. 265, Drift rate is associated with left superior parietal thickness in mutation carriers

L. 274, Left superior parietal thickness presents a bell shape pattern related to age-at-onset

L. 288, Response threshold is correlated with hippocampus volume in preHDs

- The linear model parameters (not only the t values) should be reported in order to understand the strength of the relationship between drift rate and cortical thickness. Particularly in the second paragraph of the section.

We thank the reviewer and apologize for this oversight. The estimates are now included in the manuscript (l. 259-262).

A higher volume of the hippocampus was associated with a higher drift rate only in preHDs ($\beta=2246 \pm 555$, 95%CI [1118, 3374], $t(33)=4.05$, 95%CI [948, 3543], $p<.001$) and not in earlyHDs ($\beta=-160.3 \pm 351$, 95%CI [-874, 553], $t(33)=-0.46$, 95%CI [-9890, 660], $p=.88$)

a higher volume of the hippocampus predicted a higher response threshold in preHDs ($\beta=2163.5 \pm 892$, 95%CI [349, 3978], $t(33)=2.432$, 95%CI [77, 4250], $p<.05$), but not in earlyHDs ($\beta=-10.9 \pm 564$, 95%CI [-1158, 1137], $t(33)=-0.02$, 95%CI [-1330, 1309], $p=.99$).

Minor:

- p.6 2nd paragraph, what does “least” next to the p-value mean?

“least” means that the value is at least inferior to the value indicated.

- p.7 “On the contrary” statistics are missing in the main text for this paragraph

All statistics are included in Table 3. We chose not to include them in the text to ease reading and avoid redundancy.

- p.9 first paragraph, Brain reserve should be introduced in the introduction as this seems to be an interesting and important concept in this research.

We thank the reviewer for this suggestion and explain better in the first paragraph of the introduction the different concepts of cognitive reserve, neural reserve, neural compensation and brain reserve (l. 44-53).

Neurodegenerative diseases affect brain parts and functions at variable degrees and at different stages over the course of the disease, and eventually precipitate brain atrophy that precedes intellectual deterioration¹. In general, patient’s normal behavior is maintained until the neuropathological damage surpasses the adaptive capabilities of the brain leading to the appearance of clinical symptoms²⁻⁴. The concept of “reserve” refers to this capacity of the brain to resist neuropathological changes and preserve cognitive functioning⁵. Reserve is thought to rely on brain reserve and cognitive reserve. While brain reserve relies purely on quantitative aspects of the brain such as brain size for example, cognitive reserve reflects the brain’s capabilities to optimize and develop alternative cognitive strategies to actively preserve cognitive functions. Cognitive reserve depends on patient’s lifetime intellectual activities and environmental factors. It relies on two concepts: neural reserve and neural compensation. The brain can either increase the efficiency of an existing yet deteriorating network (neural reserve) and/or recruit other regions upon performing a task (neural compensation)^{3,6}; i.e., some cognitive functions may compensate for others that were impacted at earlier stages.

- The description of the DDM fitting strategy should be in the method section rather than in the supplementary material.

We have moved the description of the DDM fitting strategy to the Methods section (l. 597-629)

- Appendix

-- Supplementary Table 1: the leading zeros makes this table hard to read, maybe use a log base 10 transformation to make it more easily readable

We agree that the leading zeros are difficult to read. But in this table, the values are statistical results, standard deviations and confidence intervals. We are afraid that the transformation into a logarithmic base will make them difficult to interpret. To make the table more readable, all zeros before the decimal point have been removed.

-- Supplementary Figure 1. hard to see, maybe put the figure row-wise instead of column-wise to be able to better see the areas presenting cortical thickness differences.

In the new Figure, the two panels are now in column.

Supplementary Figure 1. Neuroanatomical differences between groups

(A) Cortical maps showing significant cortex thinning in earlyHDs as compared with imaging controls. Each cluster is represented in a different color: blue: left angular gyrus, purple: left occipital superior, orange: right lateral occipital, and yellow: right caudal middle frontal. (B) Cortical maps of differences between earlyHDs and preHDs showing significant thinner cortex in earlyHDs compared with that of preHDs. Yellow represents right lateral occipital, light grey represents gyrus, and dark grey represents sulcus.

-- Model selection refers to Table 2 while it should probably be Table 5
This is now corrected.

Gabriel Weindel

Reviewer #3 (Remarks to the Author):

This study collected behavioural data from a binary auditory discrimination task from earlyHDs, preHDs and controls. EarlyHDs had lower accuracy and slower RT than the rest two groups, and preHDs had comparable performance as controls. The authors fitted DDMs to the behavioural data. Group differences in behavioural performance yielded lower decision thresholds in earlyHDs (compared with preHDs and controls). On drift rate, it is the lowest in the earlyHDs. Further subgroup analysis indicated that among preHDs, the drift rates differed according to their time to predicted age-at-onset, for which the authors interpreted as a compensatory mechanism. The manuscript is in

general easy to follow. However, the authors need to consider the following issues in their analyses and results.

1. Cognitive compensation. The manuscript reported no significant difference in behavioural measures between preHDs and controls. Throughout the manuscript (Intro, results and discussion), the lack of behavioural difference together with fitted DDM parameters were used to support a compensation hypothesis: drift rate was increased to compensate for the decrease in decision threshold (e.g., line 301). However, I feel this proposition does not offer new insights (in terms of cognitive compensation) beyond its immediate interpretation of the data, and, in fact, the data cannot be adequately summarised by this statement. First, a larger decision threshold in earlyHDs vs controls does not necessarily imply a decision deficit but can simply be a more conservative trade-off towards decision accuracy over speed in earlyHDs.

We thank the reviewer for pointing out that this point was not clear. We agree with the reviewer that a larger decision threshold in earlyHDs vs controls does not necessarily imply a decision deficit. Yet, in a case of a conservative trade-off towards decision accuracy, the increase in threshold should be related to better accuracy. This is not the profile of responses we observed in the data. Indeed, analyses revealed that the earlyHDs are less accurate and slower than controls, showing a performance impairment compared to controls. We clarified this point in the paragraph in the discussion (l. 354-357)

Provided that earlyHDs' other cognitive disabilities are not too severe, longer decision times should improve their accuracy (speed-accuracy tradeoff)⁴³. Yet, this is not the profile of responses we observed as earlyHDs are less accurate and slower than controls, showing a performance impairment compared to controls.

Second, relating to this, the increase in HD's decision threshold in HDs does not mean more impulsive decisions, opposite to the claim in the manuscript.

We thank the reviewer for pointing out this typo and corrected (l. 346-348)

Disruption of that indirect pathway, and a decrease in the number of white matter fibers extending between the striatum and the cortex^{38,39} in Huntington's disease should decrease the inhibition of the subthalamic nucleus and lead to less impulsive choices and an increase in response threshold⁴⁰

Third, there is no group difference in the threshold between preHDs and controls, and the manuscript did not report any bell shaped parameter values from different preHD subgroups.

Focusing on decision threshold, a bell-shaped pattern would have resulted in a threshold value for controls lying between the value for preHDs and earlyHDs. This is not the pattern we observed for threshold values. Indeed, decision threshold analysis showed a progression from lowest threshold in controls to highest in early HDs, with preHD values in between, without any bell shape to report. This is reported l.205-207.

Analyses showed a significant difference between groups in terms of the response threshold ($F(2,90)=4.79$, $p<.05$) and the drift rate ($F(2,90)=5.60$, $p<.01$), unlike non-decision time ($F(2,90)=0.13$, $p=.88$) and relative bias ($F(2,90)=0.24$, $p=.79$). Tukey's post-hoc showed progression of the response threshold from the lowest in controls to the highest in earlyHDs, with the values of the preHDs in between. This suggests an increase of the response threshold over the course of the disease (Table 3, Fig. 3A).

Together, I felt that the current study does report some interesting findings of structural correlates of the drift rate, but the results do not provide clear support for a compensatory mechanism.

We thank the reviewer for pointing out that the link between DDMs parameter and cognitive reserve was not sufficiently clear. We have completely rewritten the introduction (paragraph 1 and 3), explaining in more detail why DDMs can be useful to investigate the mechanisms of cognitive reserve separately from cognitive dysfunction, and how these models, which differentiate the cognitive parameters that lead to behavioral performance, give us the opportunity to distinguish between deficient parameters and compensatory parameters.

We have also rewritten the paragraph explaining DDM in Results section.

Finally, in the discussion we interpret these findings as a possible explanation of cognitive reserve with caution and point out limitations and future directions.

Introduction (I.44-53)

*Neurodegenerative diseases affect brain parts and functions at variable degrees and at different stages over the course of the disease, and eventually precipitate brain atrophy that precedes intellectual deterioration*¹. In general, patient's normal behavior is maintained until the neuropathological damage surpasses the adaptive capabilities of the brain leading to the appearance of clinical symptoms²⁻⁴. The concept of "reserve" refers to this capacity of the brain to resist neuropathological changes and preserve cognitive functioning. Reserve is thought to rely on brain reserve and cognitive reserve. While brain reserve relies purely on quantitative aspects of the brain such as brain size for example, cognitive reserve reflects the brain's capabilities to optimize and develop alternative cognitive strategies to actively preserve cognitive functions. Cognitive reserve depends on patient's lifetime intellectual activities and environmental factors. It relies on two concepts: neural reserve and neural compensation. The brain can either increase the efficiency of an existing yet deteriorating network (neural reserve) and/or recruit other regions upon performing a task (neural compensation)^{3,5}; i.e., some cognitive functions may compensate for others that were impacted at earlier stages.

I.70-84

To study the mechanisms of cognitive reserve separately from cognitive dysfunction¹⁰⁻¹³, we selected a cognitive task that is impaired in the early stages of the disease (no longer effective cognitive reserve) but still normal in presymptomatic subjects (effective cognitive reserve). We used a language discrimination task as language is one of the first cognitive function to decline in Huntington Disease¹³⁻¹⁸, with normal or near-normal performance in preHDs and abnormal performance in earlyHDs, suggesting that language performance would be a reliable measure to study cognitive reserve. We analyzed the results of the language task - deciding whether two items are similar or not - through Drift diffusion models (DDMs)^{19,20}. These models enable us to evaluate separately the cognitive parameters involved in the discrimination task: accumulation over time of sensory evidence at a certain rate up to a response threshold that triggers the motor response to indicate which of the two alternatives to pick (similar or not). The hypothesis is that in preHD, one of the impaired cognitive parameters will be compensated for by another cognitive parameter, resulting in normal behavioral performance, whereas in earlyHD, the compensatory cognitive parameter will no longer be effective, resulting in a behavioral deficit.

Results section, I. 228-239

The participant's behavioural performance is the result of the drift rate and response threshold parameters. Drift rate appears to show a compensatory pattern (faster rate of evidence accumulation) while response threshold shows a deteriorative pattern (higher response threshold) throughout all stages of the disease. We went further and investigated the relationship between drift rate and response threshold. We fitted a linear model with the drift rate as a dependent variable, response threshold and group as predictors, and age as a covariate. This showed that an increase in response threshold was associated with an increase in drift rate. The increase was similar in controls and earlyHDs, and sharper in preHDs (Table 4, Fig. 3E), meaning that for preHDs the drift rate might compensate for the increasing threshold allowing to preserve behavioral performances. This relationship between parameters was not induced by contamination between them as they were all not correlated (all $p > .05$)²⁶.

Discussion, l.407-416

Such an increase in cortical thickness, supposedly caused by hyperactivation, was associated with better performances (shorter response times, better accuracy, and higher drift rates) (Supplementary Fig. 2, Supplementary Table 4, Fig. 4D,E), supporting the hypothesis of a successful compensation, as previously reported in motor learning²⁹. The post-hoc analysis showed that only preHDs middle group had this cortical hypertrophy, not preHDs far from and close to predicted age-at-onset; a bell-shaped pattern in favor of a compensatory mechanism emerging at a certain point and failing as the pathological load increases. Although these findings would need to be replicated in a larger cohort, they are consistent with the results of previous studies on Alzheimer's disease⁵³ and Huntington's disease⁵⁴ which reported a preclinical stage of hypertrophy and increased functional connectivity between the left caudate nucleus and parietal lobe preceding atrophy in symptomatic patients³². This may reflect an experience-dependent increase in neural volume^{55,56} as an attempt to compensate for the dysregulation of the striatal network.

2. Models and statistics. First, the model assumed the same absolute drift rate value between "same" and "different" choices. Please provide evidence to support this model design decision. For example, could the author confirm that there was no difference in any behavioural measures between the two choices (e.g., Bayesian statistics supporting the null is appropriate here)?

We thank the reviewer for the opportunity to clarify the reason for this choice. As described in the manuscript, we used stimulus coding rather than accuracy coding in our model in order to be able to take into account a possible bias toward answering "same" or "different". It is not possible mathematically to estimate at the same time a difference in drift and a possible bias (see <https://hddm.readthedocs.io/en/latest/howto.html#code-subject-responses>). We clarified the manuscript as follow : (l. 588-589)

We assumed the same absolute drift rate value for both answers ("same" and "different"), a necessary hypothesis to estimate a possible relative bias toward one of the answers.

That being said, we ran bayesian paired t-tests to compare accuracy and reaction time for "same" and "different" trials. We found no differences between trial types in any measures and Bayes factor in favor of the null hypothesis for both accuracy ($BF_{01} = 8.4$) and reaction time ($BF_{01} = 6.5$).

Second, the manuscript opted for the "parsimonious" model over the "full" model. Considering that the two models have very close DIC values, could the authors confirm that their main conclusions are not changed if using parameters from the full model?

We have opted for the “parsimonious” model rather than the “full” model because of 'Occam's razor', or the principle of parsimony, which states that if there is a choice between two equally suitable models, all other things being equal, it is generally preferable to choose the simpler, or more parsimonious, model. There is generally a tradeoff between goodness of fit and parsimony: low parsimony models (i.e. models with many parameters) tend to have a better fit than high parsimony models. This is not usually a good thing; adding more parameters usually results in a good model fit for the data at hand, but that same model will likely be useless for predicting other data sets. This is called over fitting.

In our manuscript the full model is less parsimonious and has a higher DIC than the parsimonious model which means that the full model fits the data worse than the parsimonious model. Given the previous points mentioned (principle of parsimony and DIC in favor of the parsimonious model), there's no argument in favor of the full model since it's less parsimonious and fits the data less well.

However, to address the concerns of the reviewer, we verified that the distinctions between groups remain consistent when analyzed with the full model. The results remained unchanged. We reproduced the dedicated part of Table 3 and incorporated the statistical data from the full model in blue.

	Descriptive statistics			Group comparisons		
	controls	preHDs	earlyHDs	preHDs/controls	earlyHDs/controls	preHDs/earlyHDs
Response threshold, a	2.37±0.73 2.35±0.72	2.8±1.00 2.73±0.96	3.02±1.18 3.06±1.2	[-0.19,1.1], p=.24 [-0.25,0.95], p=.34	[0.15,1.22], p<.01 [0.19,1.26], p<.01	[-0.93,0.38], p=.57 [-1.02,0.28], p=.36
Drift rate, v	2.64±0.82 2.63±0.81	3.13±0.96 3.09±0.94	1.54±0.47 1.54±0.47	[-0.078,0.93], p=.11 [-0.098,0.90], p=.14	[-1.48,-0.58], p<.001 [-1.47,-0.57], p<.001	[0.91,2.01], p<.001 [0.87,1.97], p<.001
Non-decision time, Ter	0.07±0.07 0.07±0.07	0.07±0.06 0.08±0.06	0.08±0.10 0.08±0.10	–	–	–
Bias, zr	0.49±0.03 0.49±0.03	0.49±0.02 0.49±0.02	0.49±0.04 0.49±0.04	–	–	–

Mean response threshold, mean drift rate, mean non-decision time and mean Bias of participants in the auditory language discrimination task. Descriptive statistics (mean±standard deviation) of each group. Group comparisons: Tukey's post-hoc test [95% confidence interval] and p-value of each pair comparison are displayed only if ANOVA showed significant main effect of group. preHDs: premanifest participants; earlyHDs: early-stage Huntington's disease patients.

Third (please correct me if I am wrong), throughout the manuscript, point estimates of individual subject's model parameters were used for frequentist statistics (e.g., ANOVA and regression). I believe such calculations are not valid, because individuals' posterior model parameters are sampled from group-level posteriors. As a result, they are no longer independent of each other and hence are not fit for subsequent frequentist statistics. A more proper method is to do a Bayesian comparison directly on the group-level posteriors. For regression, the authors should consider using regression models during MCMC (i.e., adding MRI-derived measures as regressors during HDDM model fit).

We thank the reviewer for pointing to the regression method during MCMC. Yet, running regression with the MRI data at once is technically impossible as the MRI data for the imaging control group and the DDM data from the behavioral control group do not come from the same participants.

Concerning the comparison of posterior distributions, the post-hoc division of the preHDs into three small groups does not allow running a DDM for each sub-group. Due to the number of trials and participants, such models fail to converge. Reporting posterior distribution for these groups is therefore impossible as well as performing inferences based on these posterior distributions. For the purpose of clarity and consistency in the paper, we avoided reporting different types of statistics and have instead followed a method already used in several previous studies (Zhang et al., 2026; Liu et al., 2022; Le Stanc et al., 2023). We think that this would help the understanding of the data and make the paper easier to follow.

3. Model fit. Supplementary Table 5. Please report posterior model checks for each group. In addition, the authors should clarify the use of “Proportion of “same” responses” as a posterior check metric. (1) It appears that the fitted value has very large std; and (2) It is unclear how it relates to the high accuracies observed in behavioural data.

Supplementary Table 5 was intended to provide the information about the goodness of fit but we added a graphical representation of the information by plotting the posterior predictive checks for the different groups and the two different models in Supplementary Figure 3 and added the reference to it in the main manuscript. We hope that it helps clarify the appropriateness of fit for each group.

Supplementary Figure 3. Posterior predictive checks for the full and parsimonious models The distribution along the positive x-axis shows the reaction time distribution for “different” trials in green while the distribution along the negative x-axis shows the reaction time distribution for “same” trials in yellow. Each panel shows the normalized histograms of the observed data and the model prediction (solid black line) for each group (controls, preHDs and earlyHDs).

4. PreHD subgrouping. Please provide a rationale for the subgrouping criteria used in the study. Why were <1 yr and >10 yrs used to split the group into three unequal subgroups? With N as small as =4 in one subgroup, are the authors confident that their results are not strongly susceptible to type II errors?

We thank the reviewer for pointing out that the group splitting was not sufficiently justified. We have added references in the new version of the article and acknowledged in the discussion section that the small number of subjects in each group is a limitation of our study.

Results section I. 214-220

To further investigate this pattern, we split the preHDs into three groups according to their time to predicted age-at-onset ⁷. Determination of disease onset in Huntington's disease is made by clinical experience, but the conversion is a progressive process which makes it difficult to determine the exact moment of motor onset of HD ²⁴. PreHD were thus stratified into three groups, far, middle and close to the onset of the disease, according to the time remaining until the predicted onset of the disease : close to onset with predicted onset within a year (N=4), far to disease onset with a predicted onset ≥ 10 years (N=7) and middle onset with a predicted onset between 1 to 10 years (N=9) ²⁵.

Discussion I.412-416,

Although these findings would need to be replicated in a larger cohort, they are consistent with the results of previous studies on Alzheimer's disease ⁵³ and Huntington's disease ⁵⁴ which reported a preclinical stage of hypertrophy and increased functional connectivity between the left caudate nucleus and parietal lobe preceding atrophy in symptomatic patients ³².

5. Cortical thickness analysis. I am not sure if this analysis is circular and I hope the authors can clarify. The cortical cluster was first identified as a significant correlation with the drift rate across all subjects (line 246), and the cortical thickness from the cluster was then compared between groups and preHD subgroups. Since we already know that the drift rate differs between groups, the subsequent ANOVA and t-tests are not independent of the whole-brain analysis.

We identified a region of interest using the correlation between drift rate and the whole brain of mutation carriers but not behavioral controls as they do not have imaging data. Using the identified cluster, we then compare cortical thickness of the groups to the imaging controls which are different subjects. In that sense, we believe that the comparison to controls is not circular. In addition, we are not trying to identify the strength of the relationship between drift rate and cortical thickness in this cluster which is our understanding of a circular analysis according to Kriegeskorte et al., 2010.

Another way to perform a totally independent analysis would be to perform a cross-validation of the results either by replicating or by splitting our cohort in different groups, one to identify the region of interest and the other to compare groups' cortical thickness between them. Yet, we are studying Huntington's disease. The advantage of this disease is that it is one of the only genetic models of neurodegenerative disease that makes it possible to follow patients before the onset of the disease. Unfortunately, it is a rare disease, which explains the small size of the cohort. Here, our cohort is not large enough to carry out cross-validation.

We acknowledge that the number of participants is small and that these results should be replicated in a larger cohort (see above paragraph on limitations in the discussion section).

Minor,

1. I am puzzled to understand why the behavioural task is termed a language task since only pseudo-words were used.

Language processing is made up of several elements: phonology (sounds of speech), morphology (how sounds combine to form words), lexicon (words), syntax (how words combine to form sentences) and concept (semantic knowledge). We focus here on the phonological level, which is the transformation of acoustic sounds into native phonemes. This process is the first level of language comprehension, it depends on the native language and its deficit prevents language comprehension (Franklin, 1989;

Jacquemot et al., 2003). The pseudoword discrimination task is a classical task used to assess phonological process (Jacquemot et al., 2019).

2. Line 337 Fig. 3F  Fig. 3E?

We thank the reviewer for their careful reading of the manuscript. We apologize for the typo that has been corrected in the new version of the manuscript.

3. On several occasions (e.g., line 238-239), $t(33)=xx$ was reported for different groups or preHD subgroups. Please check the DoF as they should not be all the same (given the different numbers of subjects in each group)?

These are the degrees of freedom of the linear model associated with the results. They do not change for each group. We used the function `emmeans` from the package `emmeans` in R to obtain the slopes for each group.

For example following our linear model called `s_relationship_v` showing a interaction between group and the hippocampus structure, we ran :

```
>emmeans(s_relationship_v, ~ legend, var = "hippocampus")
>test(emmeans(s_relationship_v, ~ legend, var = "hippocampus"), adjust = "tukey")
```

Here is the output :

legend	hippocampus.trend	SE	df	lower.CL	upper.CL
earlyHDs	-160	351	33	-874	553
preHDs	2246	555	33	1118	3374

Confidence level used: 0.95

legend	hippocampus.trend	SE	df	t.ratio	p.value
earlyHDs	-160	351	33	-0.457	0.8779
preHDs	2246	555	33	4.050	0.0006

P value adjustment: sidak method for 2 tests

To clarify that these statistics refer to the output of a linear model, and to give an idea of the strength of the relationship, we modified the manuscript and reported the slopes.

I. 260-262

A higher volume of the hippocampus was associated with a higher drift rate only in preHDs ($\beta=2246 \pm 555$, 95%CI [1118, 3374], $t(33)=4.05$, 95%CI [948, 3543], $p<.001$) and not in earlyHDs ($\beta=-160.3 \pm 351$, 95%CI [-874, 553], $t(33)=-0.46$, 95%CI [-9890, 660], $p=.88$)

I. 290-292

a higher volume of the hippocampus predicted a higher response threshold in preHDs ($\beta=2163.5 \pm 892$, 95%CI [349, 3978], $t(33)=2.432$, 95%CI [77, 4250], $p<.05$), but not in earlyHDs ($\beta=-10.9 \pm 564$, 95%CI [-1158, 1137], $t(33)=-0.02$, 95%CI [-1330, 1309], $p=.99$).

References

Barulli, D., and Stern, Y. (2013). Efficiency, capacity, compensation, maintenance, plasticity: Emerging concepts in cognitive reserve. *Trends Cogn. Sci.* 17, 502–509. 10.1016/j.tics.2013.08.012

- Dehaene-Lambertz, G., & Baillet, S. (1998). A phonological representation in the infant brain. *Neuroreport*, 9(8), 1885-1888.
- Franklin, S. (1989). Dissociations in auditory word comprehension; evidence from nine fluent aphasic patients. *Aphasiology*, 3(3), 189-207.
- Giavazzi, M., Daland, R., Palminteri, S., Peperkamp, S., Brugieres, P., Jacquemot, C., ... & Bachoud-Lévi, A. C. (2018). The role of the striatum in linguistic selection: Evidence from Huntington's disease and computational modeling. *Cortex*, 109, 189-204
- Jacquemot, C., Pallier, C., LeBihan, D., Dehaene, S., & Dupoux, E. (2003). Phonological grammar shapes the auditory cortex: a functional magnetic resonance imaging study. *Journal of Neuroscience*, 23(29), 9541-9546.
- Jacquemot, C., Lalanne, C., Sliwinski, A., Piccinini, P., Dupoux, E., & Bachoud-Lévi, A. C. (2019). Improving language evaluation in neurological disorders: The French Core Assessment of Language Processing (CALAP). *Psychological assessment*, 31(5), 622.
- Kriegeskorte N, Lindquist MA, Nichols TE, Poldrack RA, Vul E. Everything You Never Wanted to Know about Circular Analysis, but Were Afraid to Ask. *Journal of Cerebral Blood Flow & Metabolism*. 2010;30(9):1551-1557. doi:[10.1038/jcbfm.2010.86](https://doi.org/10.1038/jcbfm.2010.86)
- Le Stanc, L., Youssov, K., Giavazzi, M., Sliwinski, A., Bachoud-Lévi, A. C., & Jacquemot, C. (2023). Language disorders in patients with striatal lesions: deciphering the role of the striatum in language performance. *Cortex*, <https://doi.org/10.1016/j.cortex.2023.04.016>.
- Liu, Wang, Wang, Xiao, Shi, The influence of reward anticipation on conflict control in children and adolescents: Evidences from hierarchical drift-diffusion model and event-related potentials, *Developmental Cognitive Neuroscience*, Volume 55, 2022, 101118, <https://doi.org/10.1016/j.dcn.2022.101118>.
- Näätänen, R., Lehtokoski, A., Lennes, M., Cheour, M., Huotilainen, M., Iivonen, A., ... & Alho, K. (1997). Language-specific phoneme representations revealed by electric and magnetic brain responses. *Nature*, 385(6615), 432-434.
- Zhang, J., Rittman, T., Nombela, C., Fois, A., Coyle-Gilchrist, I., Barker, R.A., Hughes, L.E., and Rowe, J.B. (2016). Different decision deficits impair response inhibition in progressive supranuclear palsy and Parkinson's disease. *Brain* 139, 161–173. [10.1093/brain/awv331](https://doi.org/10.1093/brain/awv331).

REVIEWERS' COMMENTS:

Reviewer #1 (Remarks to the Author):

The authors have satisfactorily answered to all my previous concerns, therefore I have no further comments.

Reviewer #2 (Remarks to the Author):

The authors responded clearly to my concerns. I still think the splitting into three preHD groups and the two-step analysis of DDM parameters is sub-optimal and might not generalize in other samples. Nevertheless, I understand the reasons evoked by the authors and see that the manuscript has been edited with the necessary caution in the interpretation of the results.

Reviewer #3 (Remarks to the Author):

The authors addressed most of my concerns in the revision. I just have one further comment.

In the revision, the authors claimed that "[...] in Huntington's disease should decrease the inhibition of the subthalamic nucleus and lead to less impulsive choices and an increase in response threshold". This does not seem to be in line with the prevalence of impulsive behaviour in HD.